# A new time-independent formulation of fractional release

Jennifer Ostermöller[1], Harald Bönisch[2], Patrick Jöckel[3], and Andreas Engel[1]

[1]Institute for Atmospheric and Environmental Sciences, Goethe University Frankfurt, 60438 Frankfurt, Germany
[2]Institute of Meteorology and Climate Research, Karlsruhe Institute of Technology, 76344 Karlsruhe, Germany
[3]Deutsches Zentrum für Luft- und Raumfahrt (DLR), Institut für Physik der Atmosphäre, Oberpfaffenhofen, Germany

*Correspondence to:* J. Ostermöller (ostermoeller@iau.uni-frankfurt.de)

**Abstract.** The fractional release factor (FRF) gives information on the amount of a halocarbon that is released at some point in the stratosphere from its source form to the inorganic form, which can harm the ozone layer through catalytic reactions. The quantity is of major importance because it directly affects the calculation of the Ozone Depletion Potential (ODP). In this context time-independent values are needed which, in particular, should be independent of the trends in the tropospheric mixing ratios (tropospheric trends) of the respective halogenated trace gases. For a given atmospheric situation, such FRF values would represent a molecular property.

We analyzed the temporal evolution of FRF from ECHAM/MESSy Atmospheric Chemistry (EMAC) model simulations for several halocarbons and nitrous oxide between 1965–2011 on different mean age levels and found that the widely used formulation of FRF yields highly time-dependent values. We show that this is caused by the way that the tropospheric trend is handled in the widely used calculation method of FRF.

Taking into account chemical loss in the calculation of stratospheric mixing ratios reduces the time-dependence in FRFs. Therefore we implemented a loss term in the formulation of FRF and applied the parameterization of a "mean arrival time" to our data set.

We find that the time-dependence in FRF can almost be compensated by applying a new trend correction in the calculation of FRF. We suggest that this new method should be used to calculate time-independent FRF, which can then be used e.g. for the calculation of ODP.

## 1 Introduction

Chlorine- and bromine-containing substances with anthropogenic sources have a strong influence on ozone depletion in the stratosphere. The gases are emitted in the troposphere, where many of them are nearly inert before they enter the stratosphere at the tropical tropopause. In the stratosphere, many of the molecules will be broken down photochemically and release halogen radicals that intensify ozone destruction (Solomon, 1990).

The fraction of a halocarbon at some point in the stratosphere that is released from the organic (source) gas into the inorganic (reactive) form is quantified by its fractional release factor (FRF). The quantity was defined by Solomon et al. (1992) as "the fraction of the halocarbon species x injected into the stratosphere that has been dissociated". It can be calculated by comparing the original mixing ratio of a tracer that entered the stratosphere to the mixing ratio that is observed at some point in the

stratosphere. The difference of this entry mixing ratio and the stratospheric mixing ratio is equal to the amount of the species released due to photochemical breakdown.

When entering the stratosphere at the tropical tropopause, ozone depleting substances (ODS) have a FRF that is zero. As they follow the stratospheric circulation, the air parcels get distributed by different transport pathways and pass through their photochemical loss regions, where the molecules get dissociated. The FRF increases until it reaches the value of 1 when the ODS is completely depleted and all halogen atoms it contained have been released.

FRF thus describes the effectiveness with which a certain ODS is broken down in the stratosphere. For the same time spent in the stratosphere, shorter lived species will have higher FRF than longer lived molecules. FRF are therefore used in the calculation of the Ozone Depletion Potential (ODP), a quantity which describes how effective a certain chemical is at destroying stratospheric ozone (Solomon et al., 1992). FRF should thus be specific for a given molecule and a given atmospheric condition. If atmospheric conditions, e.g. stratospheric dynamics or the actinic flux responsible for photochemical degradation, change, FRF is expected to change. However, FRF should not be dependent on the trend in the tropospheric mixing ratios of the chemical compound (tropospheric trend) under otherwise unchanged atmospheric conditions.

For every tracer with changing tropospheric mixing ratios, we thus need to ensure that this trend does not affect the FRF values derived from stratospheric observations. The observed mixing ratio of a chemically active species (CAS) in the stratosphere is, however, influenced by its tropospheric trend and by chemical breakdown. Only the latter should contribute to the FRF. In the calculation of FRF the tropospheric trend thus needs to be taken into account and corrected for. As the different transit pathways contributing to the air parcel are associated with different transit times and different photochemical breakdown, the complex interplay between transport, mixing and photochemistry needs to be described correctly for this purpose.

In recent years, inconsistencies between FRF values derived from independent observations at different times were identified (Laube et al., 2013; Carpenter et al., 2014). This could either be caused by real changes in FRF, due to changing atmospheric conditions, or by deficiencies in the way that the tropospheric trends are taken into account in the calculation of FRF. The latter is very likely, as data from different time periods are compared, where trends differ not only in magnitude but sometimes even in the direction (positive/ negative trend), suggesting possibly large impacts of the way that tropospheric trends are considered in the calculation of FRF.

In the current formulation for the calculation of FRF (Newman et al., 2007), transport and mixing of chemically active species are treated in a similar way as for chemically inert species, which are used to derive mean age of air. In brief this concept of mean age of the air (Hall and Plumb, 1994; Waugh and Hall, 2002) relies on the idea that different transport pathways (and associated transit times) contribute to the chemical composition of an air parcel at a given point in the stratosphere. Different transit times associated to these different transport pathways have different probabilities, which are described by a probability density function (pdf) also known as age spectrum. By folding the probability distribution for a certain entry time into the stratosphere with the time series of the inert trace gas, its mixing ratio at this point in the stratosphere can be derived, as long as there is no chemical loss. The transit time distribution is called the age spectrum $G$ and the first moment (the arithmetic mean) is called mean age $\Gamma$. Plumb et al. (1999) showed that this concept is only valid to describe the propagation of inert tracers into the stratosphere. The underlying reason is that air parcels which have already spent a lot of time in the stratosphere,

will only contribute very little to the observed mixing ratio of a compound which experiences photochemical loss, as a large fraction of the molecules of this compound will not be in the organic form anymore. Air parcels with long transit times thus need to be weighted less heavily then air parcels with short transit times. Based on that finding, Plumb et al. (1999) introduced species dependent arrival time distributions (ATD) to characterize the propagation of tracers undergoing chemical loss and with changing tropospheric abundances.

Plumb et al. (1999) used a two dimensional model to calculate the species dependent mean arrival time $\Gamma^*$, which is the first moment of the species dependent ATD. They showed that there is a large difference between $\Gamma$ and $\Gamma^*$ for CAS (in case of an inert tracer $\Gamma = \Gamma^*$) and that by using $\Gamma^*$ instead of $\Gamma$ it was possible to eliminate differences between correlations of different species observed at different times (and thus when tropospheric trends were different). Therefore it is likely that the differences in the the observed FRF could be influenced by the current calculation method which is based on mean age $\Gamma$ and not on the mean arrival time $\Gamma^*$.

In this paper, we first examine how strongly the FRF calculated using the current formulation is influenced by the tropospheric trend, using model calculations of FRF for some typical CAS. We show that the tropospheric trend has a significant impact on FRF. We then present a new improved formulation to calculate FRF which removes the impact of tropospheric trends much better. In Sect. 2 we present the classical and current calculation methods of FRF. A description of the ECHAM/MESSy Atmospheric Chemistry (EMAC) model and the simulations follows in Sect. 3. We then calculate FRF with the model data (Sect. 4) and show that the current calculation method yields time-dependent values. In Sect. 5 we derive a new formulation of FRF based on the concept of arrival time distribution. In Sect. 6 we show that the new calculation method yields results with much reduced influence from tropospheric trends. In the last section we discuss our results.

## 2   Calculation methods of FRF

The quantity of FRF was first introduced by Solomon and Albritton (1992) as "the fraction of the halocarbon species x injected into the stratosphere that has been dissociated". As explained above, FRF should be independent of the tropospheric trend of the species, but is expected to change if atmospheric conditions, especially stratospheric dynamics or photochemistry change. For the discussion of the different methods of calculating FRF, we assume that stratospheric transport is stationary in time, i.e. that the average circulation does not change with time. Under this assumption of unchanged stratospheric transport, the fractional release factor $f(r,t)$ should not change with time, and thus be $f(r)$, independent of $t$.

The substance specific FRF can then be expressed in general by the following equation

$$f(r) = \frac{\chi_{entry}(r,t) - \chi_{strat}(r,t)}{\chi_{entry}(r,t)}. \tag{1}$$

Herein, $\chi_{strat}(r,t)$ is the observed mixing ratio at the location $r$ and time $t$ in the stratosphere. This is an observable quantity, as it can be measured from balloon or aircraft samples or from satellites. It is influenced by temporal trends in the troposphere, stratospheric loss, and transport and mixing in the stratosphere. $\chi_{entry}$ is the representative average entry mixing ratio of an air parcel at location $r$ and time $t$. Although both $\chi_{strat}$ and $\chi_{entry}$ depend on time, $f$ should be a time-independent quantity thus

only depend on the location in the stratosphere. The representative average entry mixing ratio $\chi_{entry}$ should thus be derived in a way that $f$ is time-independent. To be consistent with previous work (Daniel et al., 1995; Schauffler et al., 2003), we will refer to this quantity as "entry mixing ratio" in the following.

In contrast to $\chi_{strat}$, $\chi_{entry}$ it is not an observable quantity but serves as a reference to describe the original mixing ratio of the CAS in the air parcel before photochemical breakdown. In case of a chemical compound which is in steady state between emissions into the atmosphere and atmospheric loss the tropospheric trend will be zero and $\chi_{entry}$ will just be its tropospheric mixing ratio. However, if the tropospheric mixing ratio of the trace gas changes with time, $\chi_{entry}$ must be calculated based on assumptions about stratospheric transport. As most ozone depleting substances are not in steady state but have tropospheric trends, this needs to be taken into account in calculating the entry mixing ratio $\chi_{entry}$. It is through the calculation of $\chi_{entry}$ that the time independence of FRF should be achieved.

In the first formulation of FRF suggested by Solomon and Albritton (1992), the entry mixing ratio was calculated from the tropospheric time series by estimating the time lag of the tracer mixing ratios between the troposphere and the point $r$ in the stratosphere based on mean age of air

$$\chi_{entry} = \chi_{trop}(t - \Gamma(r)), \tag{2}$$

where $\Gamma(r)$ is the mean age of air, which is the mean time elapsed since the entry of an air parcel at the tropical tropopause (Waugh and Hall, 2002; Hall and Plumb, 1994).

The concept of age of air (AOA) can be understood as follows: We consider a stratospheric air parcel that consists of infinitesimal fluid elements. An air parcel at some point in the stratosphere will consist of a nearly infinite number of such fluid elements. When entering the stratosphere, the fluid elements get distributed along different transport pathways. If we consider an air parcel at some location $r$ in the stratosphere, it will contain a mixture of fluid elements with longer and shorter transit times $t'$ depending on the pathway they took. Note that in the following we will use $t$ to denote time, whereas transit time (i.e. the time of a fluid element spent in the stratosphere) is denoted as $t'$. The distribution of the probabilities of the different transit times is called the age spectrum. It is denoted $G$. Assuming that the average stratospheric transport is stationary in time (i.e. no long term changes in stratospheric dynamics) the probability for a certain transit time $t'$ will only be a function of the location in the stratosphere, thus $G = G(r, t')$. In particular the age spectrum will then only depend on the location $r$ in the stratosphere and is not a function of time $t$. For simplicity, we will make this assumption of unchanged dynamics in the following, as FRF at a given location should be unchanged as long as the stratospheric transport is unchanged.

As the sum of the probabilities of all transit times must be unity, the integral of $G(r, t')$ over all possible transit times must be 1:

$$\int_0^\infty G(r, t') dt' = 1 \tag{3}$$

and the arithmetic mean of the distribution can be calculated by its first moment and is called the mean age of air

$$\Gamma(r) = \int_0^\infty t' G(r, t') dt'. \tag{4}$$

Mean age is not a directly observable quantity but it can be deduced from observations of passive tracers like $CO_2$ or $SF_6$ (Schmidt and Khedim, 1991; Hall and Plumb, 1994; Strunk et al., 2000; Engel et al., 2009).

As noted above, the photochemical breakdown of a chemical compound increases on average with the time the air parcel has spent in the stratosphere ($t'$). FRFs thus show compact correlations with mean age of air $\Gamma$. Newman et al. (2007) presented FRF as a function of mean age including the age spectrum in the calculation of $\chi_{entry}$. In this formulation, the entry mixing ratio for a certain mean age value is calculated by the convolution of the tropospheric time series with the age spectrum

$$\chi_{entry}(r,t) = \int_0^\infty \chi_{trop}(t-t')G(r,t')dt', \tag{5}$$

where $\chi_{trop}(t-t')$ represents the tropospheric mixing ratio at time $t-t'$, thus $t'$ before the date of observation. In this representation of the entry mixing ratio the transport of the species to a certain location in the stratosphere is represented by $G$. It takes into account that several transit times and pathways are possible which is an improvement compared to the representation of the entry mixing ratio according to Eq. (2) where only a single transit time is allowed for. Nevertheless, Eq. (5) is only valid for chemically inert species and does not take respect to chemical processes.

Inserting Eq. (5) into Eq. (1) yields

$$f(r) = \frac{\int_0^\infty \chi_{trop}(t-t')G(r,t')dt' - \chi_{strat}(r,t)}{\int_0^\infty \chi_{trop}(t-t')G(r,t')dt'}. \tag{6}$$

Subsequently we will refer to Eq. (6) as the "current formulation of FRF" as it has been used in Newman et al. (2007); Laube et al. (2013).

Equations (1) and (6), respectively, yield a single fractional release factor $f(r)$. In a similar way to mean age, this must be interpreted as an average value, as of course the fractional release for the fluid elements of an air parcel will differ depending in particular on the time they have spent in the stratosphere, i.e. the transit time $t'$.

The stratospheric mixing ratio $\chi_{strat}$ in Eq. (1) and Eq. (6) can be deduced from observations or from model data, as well as the tropospheric time series $\chi_{trop}$. In order to test how well the current formulation can remove the effect of tropospheric trends in the calculation of FRF, we analysed the temporal evolution of FRF using data from the EMAC model. The EMAC model and the related simulations will be presented in the next section and the time dependences of FRF calculated using Eq. (6) will be discussed in Section 4.

## 3  The EMAC Model

The ECHAM/MESSy Atmospheric Chemistry (EMAC) model is a numerical chemistry and climate simulation system that includes submodels describing tropospheric and middle atmosphere processes and their interaction with oceans, land and human influences (Jöckel et al., 2006). It uses the second version of the Modular Earth Submodel System (MESSy2) to link multi-institutional computer codes. The core atmospheric model is the 5th generation European Centre Hamburg general circulation model (ECHAM5, Roeckner et al. (2006)). For the present study we applied EMAC (ECHAM5 version 5.3.02,

MESSy version 2.51) in the T42L90MA-resolution, i.e. with a spherical truncation of T42 (corresponding to a quadratic Gaussian grid of approximately 2.8 by 2.8 degrees in latitude and longitude) with 90 vertical hybrid pressure levels up to 0.01 hPa.

## 3.1 Simulations

In this study we analyse a reference simulation performed by the Earth System Chemistry integrated Modelling (ESCiMo) initiative (Jöckel et al., 2016). The simulation RC1-base-07 is a free-running hindcast simulation from 1950 to 2011. It is forced by prescribed sea surface temperatures (SSTs) and sea ice concentrations (SICs) merged from satellite and in-situ observations. The initialization of the simulation starts in January 1950 and is followed by a spin-up period of 10 years. Therefore we will analyze the data after 1965.

The model uses observed surface mixing ratios for boundary conditions that were taken from the Advanced Global Atmospheric Gases Experiment (AGAGE, http://agage.eas.gatech.edu) and the National Oceanic and Atmospheric Administration/ Earth System Research Laboratory (NOAA/ESRL, http://www.esrl.noaa.gov).

An important point in the model set-up is the additional implementation of idealized tracers with mixing ratios relaxed to $\chi_{trop} = 1$ ppt in the lowest model layer above the surface. These idealized tracers have no tropospheric trend but the chemical kinetics in the stratosphere follow the same mechanisms as for realistic tracers. However, there is no feedback of these tracers into the chemistry, radiation or dynamics of the model. For all tracers, the chemistry is controlled by the submodel MECCA (Module Efficiently Calculating the Chemistry of the Atmosphere, Sander et al. (2011)) and the photolysis rate coefficients are calculated by the submodel JVAL (Sander et al., 2014).

Idealized tracers with constant tropospheric mixing ratios have been implemented for the halocarbons CFC-11 ($CFCl_3$), CFC-12 ($CF_2Cl_2$), methyl chloroform ($CH_3CCl_3$), Halon 1211 ($CF_2ClBr$) and Halon 1301 ($CF_3Br$), as well as for nitrous oxide ($N_2O$). These tracers have different lifetimes in the stratosphere: CFC-12 and nitrous oxide are long-lived with a similar stratospheric lifetime of 95.5 yr and 116 yr respectively (Chipperfield et al., 2013). In contrast, CFC-11 ($CFCl_3$) and methyl chloroform ($CH_3CCl_3$) are shorter-lived with stratospheric lifetimes of 57 yr and 37.7 yr respectively (Chipperfield et al., 2013). The halons have stratospheric lifetimes of 33.5 yr (Halon 1211) and 73.5 yr (Halon 1301) (Chipperfield et al., 2013). A detailed description of ECHAM/ MESSy development cycle 2 can be found in Jöckel et al. (2016), and references therein.

## 4 Time-dependence of FRF in EMAC simulations

FRFs are often analyzed as a function of mean age of air $\Gamma$ (Schauffler et al., 2003). In EMAC, the age of air is calculated from a diagnostic tracer. This tracer is linearly increasing in the lowest model layer.

To calculate FRFs according to the current formulation, we need to solve Eq. (6) and make some assumptions on the tropospheric time series and the shape of the age spectrum. For the calculation of the entry mixing ratio in the current FRF formulation (cf. Eq. (6)), Eq. (5) is integrated 30 years back in time. This is necessary to correct for the influence of the troposphere on the stratosphere. The tropospheric time series before 1950 can be taken from the RCP6.0 scenario (Meinshausen

et al., 2011). For most of the tracers considered here, the mixing ratio before 1950 was close to zero, except for the nearly linearly increasing tracer nitrous oxide (N$_2$O). N$_2$O has increased very slowly and nearly constantly by 0.8 ppb/ yr over the past decades. The tropospheric mixing ratios of N$_2$O before 1950 are assumed to decrease by the same magnitude.

In this study we use an inverse Gaussian function for the transit time distribution $G$ (Waugh and Hall, 2002; Schauffler et al., 2003) with a constant ratio of the squared width to mean age of $\Delta^2/\Gamma = 0.7$ according to Hall and Plumb (1994) and as used in previous studies (Engel et al., 2002; Bönisch et al., 2009). This parameterization can be used throughout most of the stratosphere, but varies between stratospheric models (Waugh and Hall, 2002).

As an example, Fig. 1 shows the correlations of the FRF of nitrous oxide and methyl chloroform with mean age of air using monthly mean EMAC model data and the current calculation method (cf. Eq. (6)). The correlations are compact, but not time-independent. Especially for methyl chloroform there are large differences in the correlations depending on the year. This is a first hint that there is a time-dependence in the current representation of FRF.

There may be several reasons for this time-dependency. On the one hand, changes in the stratospheric circulation or chemistry could cause changes in fractional release on a given age-isosurface, on the other hand it is possible that the tropospheric trend of the species has an impact on the derived fractional release factor.

In order to separate the two possible effects from each other, we make use of the idealized tracers described in Sect. 3.1. These tracers have nearly constant mixing ratios of 1 ppt throughout the troposphere, but in the stratosphere they experience the same transport and chemical depletion mechanisms as the realistic tracers. The FRF of the idealized tracers can easily be calculated by Eq. (1) with $\chi_{i,entry} = 1$ ppt. FRF calculated by the idealized tracers gives a very good proxy of a quasi steady state value of FRF in the model.

Assuming that the age spectra for different locations with the same mean age are similar, we investigate changes of FRF in the model on age isosurfaces instead of on geographical coordinates. As mean age e.g. at a given location shows some variability with time, this is expected to lead to reduced variability.

We calculated the temporal evolution of zonal mean FRF values derived from monthly mean data on the constant mean age of air surfaces $\Gamma = 2$ yr, 3 yr and 4 yr in the northern hemisphere mid-latitudes between 32° N and 51° N. In order to avoid possible spin-up effects, the analysis is restricted to data after 1965. The temporal evolution of FRF calculated from the idealized tracers is shown in Fig. 2.

On older mean age of air surfaces we find higher FRF values, which is reasonable, because older air has had more time to travel through the photochemical loss regions than younger air. The value of FRF depends on the species and their photolytic lifetimes. CFC-12 (CF$_2$Cl$_2$) and nitrous oxide (N$_2$O) are long-lived (cf. Sect. 3.1), even on the 4 yr age isosurface about half of the original amount remains in the organic form. In contrast, CFC-11 (CFCl$_3$) and methyl chloroform (CH$_3$CCl$_3$) are shorter-lived. These species are largely depleted on the 4 yr age isosurface with FRF values of around 0.8.

We notice a seasonality in FRF, which can be explained by seasonal variations in transport, chemistry and mixing. These are stronger in the upper stratosphere, due to shorter local lifetimes. Beside this, we can see that the FRFs for idealized tracers only slightly vary with time. The increase of FRF is in the order of about 5 % per decade, which is in agreement to Li et al. (2012), who analyzed changes in FRF in the Goddard Earth Observing System Chemistry-Climate Model (GEOSCCM). These

changes are consistent with an acceleration of the Brewer-Dobson Circulation due to climate change, which is found in EMAC calculations consistent with most other models (Butchart et al., 2010). A stronger circulation leads to a faster transport of air parcels to their loss regions and thus to an increased FRF on a given mean age level. Nevertheless, the FRF of the idealized tracers can be assumed to be a good proxy for a quasi steady state value in the model, as they are not influenced by tropospheric trends.

The temporal evolution of the FRF of the realistic tracers (with tropospheric trends) is analyzed on the same latitude band and AOA surfaces as for the idealized tracers. The results are shown in Fig. 3. The coloured lines in Fig. 3 show the results of the FRF calculation for realistic tracers according to the current formulation. The results of the idealized tracers are plotted in solid black lines and the tropospheric trends are added by dashed black lines.

It is obvious from Fig. 3 that the changes in FRF calculated for the realistic tracers are much larger than for the idealized tracers. The variation in the idealized tracers reflects the changes due to changing chemistry andy dynamics. As the only difference between the idealized and the realistic tracers is the tropospheric trend of the realistic tracers, the larger variability of FRF for the realistic tracers must be due to the way that the tropospheric trend is considered in the calculation of FRF according to the current formulation.

The results differ depending on the magnitude and on the direction of the tropospheric trend.

For $N_2O$, which has a very small linear tropospheric trend of about only 0.2 %/yr, the realistic and the idealized tracer are in good agreement, which means that the current formulation of FRF works well as long as the trends are small.

The situation is different, if we consider the anthropogenically emitted chlorofluorocarbons and methyl chloroform, which had strong positive trends in the 1980s (growth rate about 6 % for CFC-11 and CFC-12, 8.7 % for methyl chloroform (Gammon et al., 1985)) and phased out in the 1990s due to the Montreal Protocol. For those tracers, the FRF is strongly time-dependent and deviates systematically from the FRF of the idealized tracers: In times of positive trends (before 1995), FRF is underestimated in comparison to the idealized tracer. For methyl chloroform, whose positive trend is followed by a strong negative trend since the middle of the 1990s, we notice that the FRF is overestimated during the period of the negative trend compared to the idealized tracer. The chlorofluorocarbons CFC-11 and CFC-12 have a much weaker negative trend since the mid 1990ies than methyl chloroform due to their longer stratospheric lifetimes. Here, the FRF from the realistic and the idealized tracers are again in good agreement for the period with small trends.

To sum up, our model experiments show that the tropospheric trend influences the current FRF calculation and imposes a time-dependence. If trends are sufficiently small, as for $N_2O$ or the CFCs in the 21[st] century, the effect of the tropospheric trend is well removed. During periods of strong positive trends in tropospheric mixing ratios, there is a low bias in the FRF derived using Eq. (6) in comparison to the idealized tracers. During periods of strong negative trends, as observed for $CH_3CCl_3$ in the early 21[st] century the FRF based on Eq. (6) is overestimated. This time-dependence could also explain the differences between FRF values deduced from measurements at different dates. If we for instance compare CFC-12 data on the 3 yr isosurface in 1980 and in 2000, there is an increase of about 50 % on the FRF value (see Fig. 3). The result of the calculation cannot be regarded as a steady state value and the possible change due to variations in the stratospheric circulation cannot explain this

magnitude of the difference (see idealized tracers). Therefore, we conclude that it is caused by an incomplete correction of tropospheric trends and develop a new formulation of FRF in the following section.

## 5   A new formulation of FRF

As shown in Sect. 4, the currently used formulation to derive FRF does not correct for tropospheric trends in a satisfactory manner. In this Section we will show a possible reason and solution to this issue.

We consider the propagation of a CAS with solely tropospheric sources into the stratosphere. Air parcels enter the stratosphere at the tropical tropopause. In the stratosphere, the CAS gets distributed by the meridional overturning circulation (Brewer-Dobson circulation), which includes residual circulation and mixing in a similar way as for an inert tracer. In addition, the CAS will also be chemically depleted by sunlight or radicals during the transport. The mixing ratio of the CAS at a certain location in the stratosphere is thus influenced by the temporal trend in the troposphere, transport and mixing in the stratosphere, as well as loss processes. As in Sect. 2, we again make the assumption of stationary stratospheric transport, i.e. we derive a formulation of FRF which should be constant as long as stratospheric transport (and radiation) is unchanged.

In general, a tracers stratospheric mixing ratio $\chi_{strat}(r,t)$ can be formulated via its fractional release factor $f$ if we consider that it is the remaining fraction of the tracer which is not yet dissociated. This fraction $f$ will be a function of the transit pathway and the transit time $t'$. For simplification, we assume that longer transit pathways will be linked with more chemical loss and longer transit times, thus we consider $f$ to be a function of $t'$ and location $r$ only.

The mixing ratio of a chemically active substance at some point $r$ in the stratosphere at some time $t$, $\chi_{strat}(r,t)$, can be calculated by convoluting three functions: the tropospheric time series $\chi_{trop}(t-t')$, the remaining fraction due to photochemical loss $(1-f(r,t'))$, and the transit time distribution or age spectrum $G(r,t')$, which is a function of transit time $t'$ and the location in the stratosphere $r$. As explained in Sect. 2, $G$ and $f$ are not functions of time $t$ as stratospheric transport is assumed to be stationary in time.

All of the three functions depend on transit time $t'$:

$$\chi_{strat}(r,t) = \int_0^\infty \chi_{trop}(t-t')\left(1-f(r,t')\right)G(r,t')dt'. \tag{7}$$

Physically Eq. (7) describes that the observed mixing ratio of a CAS will be the sum over the mixing ratios of the individual fluid elements with different transit times, different photochemical loss and different original mixing ratios upon entry into the stratosphere. For short lived species, the fluid elements with long transit times will contribute very little to the observed mixing ratio in the stratosphere, as the original content has been photochemically depleted. The tropospheric mixing ratio at that time is thus not very relevant for the observed mixing ratio. Imagine that a CAS has a decreasing trend in the troposphere and that its fractional loss will be nearly complete after a transit time of a 4 years. The observed mixing ratio on the 4 year iso agesurface will then be dominated by the short fraction of the transit time distribution, whereas longer transit times must be weighted less heavily. The probability density function describing how strongly which transit time and thus the corresponding tropospheric

mixing ratio must be weighted should thus be different for species with different chemical loss and in particular also different for species with chemical loss then for species without chemical loss.

In the case of an inert tracer, $f(r,t') = 0$ for all possible transit times and transport pathways. Thus the loss term $(1 - f(r,t'))$ disappears and the right hand side of Eq. (7) reduces to Eq. (5). In this case the stratospheric mixing ratio $\chi_{strat}$ is identical to the entry value, as there is no chemical breakdown.

However, only if $f$ were constant for all fluid elements reaching point $r$ and independent of transit time $t'$ thus $f = f(r)$ instead of $f(r,t')$, the factor $(1 - f(r))$ could be extracted from the integral, yielding

$$\chi_{strat}(r,t) = (1 - f(r)) \int_0^\infty \chi_{trop}(t - t') G(r,t') dt', \tag{8}$$

which can be rearranged to Eq. (6), which is the form of Eq. (1) used for the calculation of the fractional release factor according to Newman et al. (2007).

As shown here, this formulation depends upon the assumption that fractional release for all fluid elements reaching point $r$ is similar for all transit times $t'$, which is clearly not a valid assumption.

In order to derive a new formulation of FRF with better correction for tropospheric trends, we again take a look at the loss term in Eq. (7). $(1 - f(r,t'))$ describes the loss as a function of transit time $t'$. In general, the fraction of a species which has been released from its source gas will depend both on the transit time $t'$ and the transport pathway the air parcel has taken. However, on average $f$ will increase the longer an air parcel has stayed in the stratosphere, especially the time spent in the loss region. For simplicity we therefore assume that $f$ will only be a function of the time spent in the stratosphere and not on the pathway. The different fractional losses for different pathways are ignored in this approach, following the "average lagrangian path" concept proposed by Schoeberl et al. (2005).

Assuming that f will only depend on the transit time $t'$, we can define a new loss weighted distribution function $G^*$, which combines $G$ with the chemical loss term $(1 - f(r,t'))$:

$$G^*(r,t') \equiv (1 - f(r,t')) \, G(r,t'). \tag{9}$$

Following Plumb et al. (1999), we will refer to $G^*$ as the arrival time distribution, as it represents the distribution of arrival times of molecules, which have not been photochemically degraded.

The arrival time distribution $G^*$ is only normalized for inert tracers without chemical loss. In this case, the loss term $f(r,t')$ disappears in Eq. (9) and the arrival time distribution coincides with the age spectrum $G^* = G$.

In general $G^*$ satisfies the relation

$$\int_0^\infty G^*(r,t') dt' \leq 1. \tag{10}$$

For this reason we define a normalized arrival time distribution $G_N^*$ by normalizing $G^*$

$$G_N^*(r,t') = \frac{G^*(r,t')}{\int_0^\infty G^*(r,t') dt'} \tag{11}$$

so that

$$\int_0^\infty G_N^*(r,t')dt' = 1 \tag{12}$$

with a corresponding mean arrival time $\Gamma^*$ that can be calculated from the first moment of $G_N^*$

$$\Gamma^*(r) = \int_0^\infty t' G_N^*(r,t')dt'. \tag{13}$$

We now solve the integral over $G^*$

$$\int_0^\infty G^*(r,t')dt' = \int_0^\infty (1-f(r,t'))\,G(r,t')dt' = \int_0^\infty G(r,t')dt' - \int_0^\infty f(r,t')G(r,t')dt' = 1 - \bar{f}(r) \tag{14}$$

with $\bar{f}$ being the first moment of the probability density function of all fractional releases, thus the arithmetic mean or average fractional release

$$\bar{f}(r) \equiv \int_0^\infty f(r,t')G(r,t')dt'. \tag{15}$$

Replacing the integral in (11) with (14) yields

$$G_N^*(r,t') = \frac{G^*(r,t')}{(1-\bar{f}(r))}. \tag{16}$$

Solving (16) for $G^*$ and inserting this relation into (9) yields

$$G_N^*(r,t')\left(1-\bar{f}(r)\right) = (1-f(r,t'))\,G(r,t'), \tag{17}$$

and inserting in (7) yields

$$\chi_{strat}(r,t) = \int_0^\infty \chi_{trop}(t-t')\left(1-\bar{f}(r)\right)G_N^*(r,t')dt' = \left(1-\bar{f}(r)\right)\int_0^\infty \chi_{trop}(t-t')G_N^*(r,t')dt'. \tag{18}$$

Note that in this formulation $\bar{f}(r)$ does not depend on transit time $t'$ and can thus be extracted from the integral.

From this equation we can now calculate the mixing ratio of a chemical active tracer at any location and time in the stratosphere as long as the tropospheric time series, the new average FRF $\bar{f}$ and the arrival time distribution are known. The other way around it is possible to infer steady state FRFs $\bar{f}$ from Eq. (18).

This can be done by simply rearranging Eq. (18) and solving for $\bar{f}$

$$\bar{f}(r) = \frac{\int_0^\infty \chi_{trop}(t-t')G_N^*(r,t')dt' - \chi_{strat}(r,t)}{\int_0^\infty \chi_{trop}(t-t')G_N^*(r,t')dt'}. \tag{19}$$

We interpret $\bar{f}$ as the mean fractional release factor on a given age isosurface. It corresponds to a quasi steady state value. Of course, FRF still depends on the mean age of air, which gives information on how long the air parcel has already spent in the

stratosphere. The new mean fractional release factor $\bar{f}$ should be independent of tropospheric trends and is only expected to change if stratospheric transport or photochemistry change. Eq. (19) is similar to Eq. (6), suggested by Newman et al. (2007), but $G$ has been replaced by the normalized arrival time distribution $G_N^*$. Note that for a species without tropospheric trend, Eq. (6) and Eq. (19) will give the identical result, as the integrals will yield the constant tropospheric mixing ratios.

The entry mixing ratio in this new formulation

$$\chi_{entry}(r,t) = \int\limits_0^\infty \chi_{trop}(t-t')G_N^*(r,t')dt' \tag{20}$$

now takes into account transport as well as chemical loss processes. Using $G_N^*$ instead of $G$ results in a lesser weighting of the tail of the transit time distribution which is reasonable, especially for CAS with short lifetimes. A shorter-lived species is almost completely depleted after a transit time of e.g. 4 years thus this transit time $t'$ should not contribute in the convolution

with the tropospheric time series when calculating the remaining organic fraction. For such shorter lived species the remaining amount in the original organic form is thus hardly influenced by the tropospheric mixing ratios of air which entered a long time ago (the "tail" of the age spectrum for an inert trace gas). The shorter lived the trace gas is, the more the weighting needs to be shifted to the short fraction of the age spectrum. The arrival time distribution describes the relevant weighting of the different transit times and is specific for each trace gas.

A complication is of course that the normalized arrival time distribution $G_N^*(r,t)$ needs to be known in order to solve Eq. (19). This arrival time distribution has been calculated from a 2D chemical transport model by Plumb et al. (1999).

Following Plumb et al. (1999), we call the first moment of this distribution the "mean arrival time" $\Gamma^*$ which takes into account the chemical loss of the species. A possible parameterization of $\Gamma^*$ was described by Plumb et al. (1999). $\Gamma^*$ is a substance specific quantity and depends on mean age and the stratospheric lifetime of the tracers. In the following section we

test the new formulation of FRF $\bar{f}$ given in Eq. (19) by applying it to EMAC model data. We compare the results to the current formulation of FRF $f$, based on Eq. (6).

## 6    Results of the new formulation

In the last section a new formulation of FRF has been derived which should be able to correct the effect of tropospheric trends when calculating FRF. We apply our new formulation Eq. (19) which takes into account effects of chemical loss to the same

data set as for the analysis of the current FRF formulation presented in Sect. 4. This means we examine the temporal evolution of FRF on the same latitude band and age of air isosurfaces again using both idealized and realistic tracers.

To solve Eq. (19) it is necessary to find a good description of $G_N^*$. We choose $G_N^*$ to have the same shape as $G$, i.e. an inverse Gaussian distribution but with the parameters $\Gamma^*$ (first moment) and $\Delta^*$ (second moment), so that $G_N^* = G(\Gamma^*, \Delta^*, t')$. Like for $G$ we use a constant ratio of the squared width to mean age of $\Delta^{*2}/\Gamma^* = 0.7$ yr.

Plumb et al. (1999) derived a parameterization of a species dependent "mean arrival time" $\Gamma^*$ for a wide range of chemically active species from a delta pulse emission calculation. $\Gamma^*$ can be calculated from the mean age of air $\Gamma$ and the mean stratospheric lifetime $\tau$ by a parameterization scheme (Plumb et al., 1999). Using $\Gamma^*$ instead of $\Gamma$ takes respect to the chemical loss

occurring on the transport pathways. We computed $\Gamma^*$ for the considered species and applied it as the first moment of our new arrival time distribution $G_N^*$.

The result of the new calculation of FRF according to Eq. (19) can be seen in Fig. 4. We clearly notice the improvement of the new calculation method. The tropospheric trend of the species is almost corrected for and FRF values for the idealized and the realistic tracers show a much better agreement.

In contrast to the current formulation (cf. Fig. 3), FRF is slightly overestimated compared to the idealized tracer in times of positive trends for CFC-12 ($CF_2Cl_2$) and methyl chloroform ($CH_3CCl_3$). For CFC-11 ($CFCl_3$), the FRF according to the new formulation is somewhat underestimated on the 2 and 3 yr age isosurface but fits the idealized tracer well on the 4yr age isosurface. Furthermore, the FRF of methyl chloroform is underestimated when tropospheric mixing ratios are declining. The reason for this feature is a too large correction between $\Gamma$ and $\Gamma^*$.

As we would expect, the fractional release of $N_2O$ is nearly unaffected by the new calculation method, because of its small tropospheric trend. For CFC-11 and CFC-12 there are still small deviations between the realistic and the idealized tracer but the steady state value is reached much earlier than in the current formulation and overall the differences are much smaller. Indeed, we do expect species and age dependent differences in the results, as the same parameterization is used to derive $\Gamma^*$ from $\Gamma$ for all mean age values and different parameters are used for different species.

The largest change can be seen for methyl chloroform, which is the analyzed substance with the largest variation in the tropospheric trend. The realistic tracer now approaches the idealized tracer and we can see the improvement especially for the highest considered age isosurface ($\Gamma = 4$ yr) in comparison to the current formulation of FRF used in Fig. 3.

To sum up, we conclude that including chemical loss into the calculation reduces the time-dependence of the FRF value substantially. The parameterization of loss was adopted from Plumb et al. (1999) who derived the parameterization from a simple 2D model. It could still be improved to obtain an even better adaption to the idealized tracer. Besides this, we also kept an inverse Gaussian distribution with a similar parameterization as for mean age, which might not be the optimal choice for the new arrival time distribution.

## 7   Summary and discussion

In this paper we presented a study on fractional release factors (FRF) and their time-dependence. We analyzed the temporal evolution of FRF between 1965 and 2011 for the halocarbons CFC-11, CFC-12 and methyl chloroform, as well as for nitrous oxide. FRF is often treated to be a steady state quantity, which is a necessary assumption to use it in the calculation of ODP and EESC. In the current formulation of FRF, the transit time distribution and the tropospheric time series of the substances are taken into account, but the coupling between trends, chemical loss and transit time distribution is not included.

For chemical active species, the fraction of the air with very long transit times (the "tail" of the transit time distribution) will have passed the chemical loss region and therefore only contributes very little to the remaining organic fraction, but is to a large degree in the inorganic form. On the other hand, the fraction of the air with short transit times will be to a large degree still in the form of the organic source gas, as it has not been transported to the chemical loss region. This must be taken into

account when folding the transit time distribution with the tropospheric time series to derive the fraction still residing in the organic (source) form. For this we used an arrival time distribution, based on the concept and parameterization suggested by Plumb et al. (1999).

We applied the two FRF calculation methods (current and new) on EMAC model data and studied the differences. For both methods we used exemplarily (but without loss of generality) zonally averaged monthly mean stratospheric mixing ratios in a latitude band between 32° N and 51° N.

A special feature of the used model simulation are the implemented idealized tracers with nearly constant tropospheric mixing ratios. We showed that the use of the new formulation of the propagation of chemical active species with tropospheric trends into the stratosphere results in FRF values, which are to a large degree independent of the tropospheric trend of the respective trace gas and thus gives a quasi steady state value of FRF. This is shown by a much better agreement with the FRF of the idealized tracers, which have no tropospheric trend.

In contrast, the classical approach yields FRF values that depend on tropospheric trends, which change with time. This might be an explanation for the discrepancies between FRF values deduced from observations at different dates. The reason for the non steady behavior is obviously based on an incomplete trend correction. In times of strong tropospheric trends, the realistic tracers deviate most from the idealized tracers. On the other hand, the FRF of the realistic $N_2O$ tracer hardly differs from the idealized tracer, because it has a very small tropospheric trend.

This may lead to discrepancies in fractional release factors derived during different time periods. Such differences in FRF have been observed between the work of Laube et al. (2013) and Newman et al. (2007). The FRF values derived by Laube et al. (2013) were lower than those derived by Newman et al. (2007) on the 3 year mean age isosurface. As the tropospheric trends were lower during the observations used by Laube et al. (2013), it is expected that the re-calculation using our method should even increase the observed difference. We therefore conclude that the calculation of mean age may be the reason for the observed discrepancies, as suggested by Laube et al. (2013).

We also acknowledge that the new formulation is less intuitive than the formulation used by Newman et al. (2007) and Laube et al. (2013). However, as we have shown that the method used by Laube et al. (2013) and Newman et al. (2007) yields values which are strongly influenced by the tropospheric trend, this loss of intuitivity and the added dependence on model information is necessary, as much more representative values are derived.

To include chemical loss into the transit time distribution, we applied the parameterization described by Plumb et al. (1999). Using the new formulation of the stratospheric mixing ratio (with loss) we constructed a new expression of the FRF and validated it with EMAC data.

The newly calculated FRF values fit well to the results of the idealized steady state tracers and the influence of the tropospheric trend can almost completely be corrected. This is remarkable, because we have to keep in mind that the parameterization was derived from a completely different and independent 2D model and that we used the same shape parameters as for the classical age spectrum.

Our new method produces FRF values which are far less dependent on tropospheric trends. In the case of small tropospheric trends the results will converge with those using the current formulation and also with those for idealized tracers without any

trends. On the other hand, more model information is needed for the derivation of the FRF values, as species dependent arrival time distributions need to be applied. The parameterisation given by Plumb et al. (1999) depends on the stratospheric lifetime of the species. As fractional release also depends on the lifetime, one may argue that there is a certain circular argumentation involved. Indeed, if the assumption on stratospheric lifetime is very far off, and tropospheric trends are large then our new

method will also fail in correcting for the tropospheric trend. However, it should be noted that the calculation is not extremely sensitive to the assumed lifetime. We investigated the sensitivity for a CFC-12 like tracer with a linearly increasing trend of 5 % per year. For an assumed steady state FRF of 0.5 at a mean age of 4 years using our method, a value of 0.5 is found with a deviation of 0.5 % for an uncertainty in the assumed lifetime of 20 %. Using the current method ignoring the effect of chemical loss would result in an FRF of 0.45, i.e. 10 % lower than the correct value. The sensitivity to the assumed lifetime is thus rather

small.

We suggest to use the new formulation and reassess former FRF data. Especially FRF values calculated from observations at times of strong tropospheric trends will profit from the new calculation method. Many fully halogenated CFCs showed strong trends prior to 1990, while many HCFCs still show very strong positive trends. This implies that FRF values currently used for HCFCs are likely to be underestimated, which would lead to an underestimation of their ODP values.

We suggest that this new method should be refined by calculating the arrival time distributions in state-of-the-art models and deriving parameterizations from these models. These new methods should be tested by including idealized tracers in the same models and subsequently be applied to observations, which have been used to derive FRF values. Using these new FRF values, a reassessment of ODP values for halogenated source gases and also a re-evaluation of temporal trends of EESC are necessary.

*Acknowledgements.* This work was supported by the DFG Research Unit 1095 (SHARP) under project number EM367/9-1 and EN367/9-2.

We thank all partners of the Earth System Chemistry integrated Modelling (ESCiMo) initiative for their support. The model simulations have been performed at the German Climate Computing Centre (DKRZ) through support from the Bundesministerium für Bildung und Forschung (BMBF). DKRZ and its scientific steering committee are gratefully acknowledged for providing the HPC and data archiving resources for this consortial project ESCiMo.

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

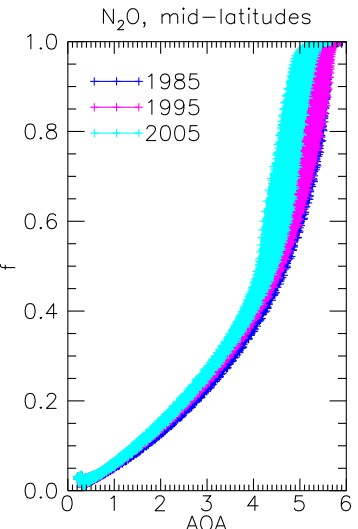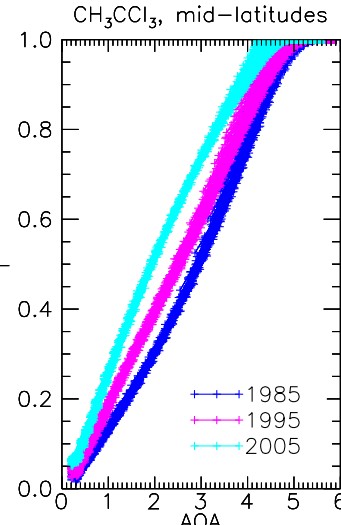

**Figure 1.** Fractional release (f) as a function of mean age of air (AOA) in the mid-latitudes between 32° N and 51° N for nitrous oxide (left) and methyl chloroform (right) derived from monthly mean EMAC model data. The FRF was calculated by the current formulation for different dates. It can be observed that the correlations vary with time.

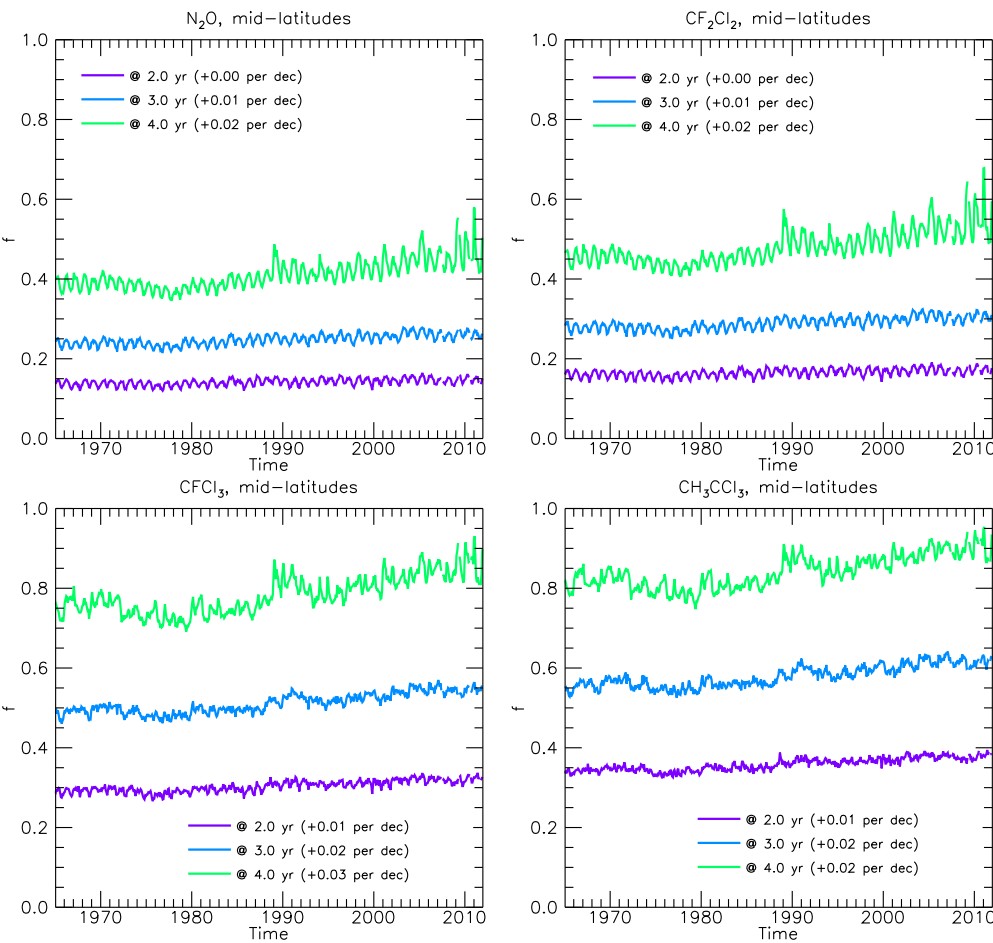

**Figure 2.** FRF calculated from the idealized tracers (without tropospheric trends) of the EMAC model in the mid-latitudes between 32° N and 51° N. The FRF is calculated on the 2 (purple), 3 (blue) and 4 yr (green) age isosurface. The absolute change of FRF per decade is noted in parentheses.

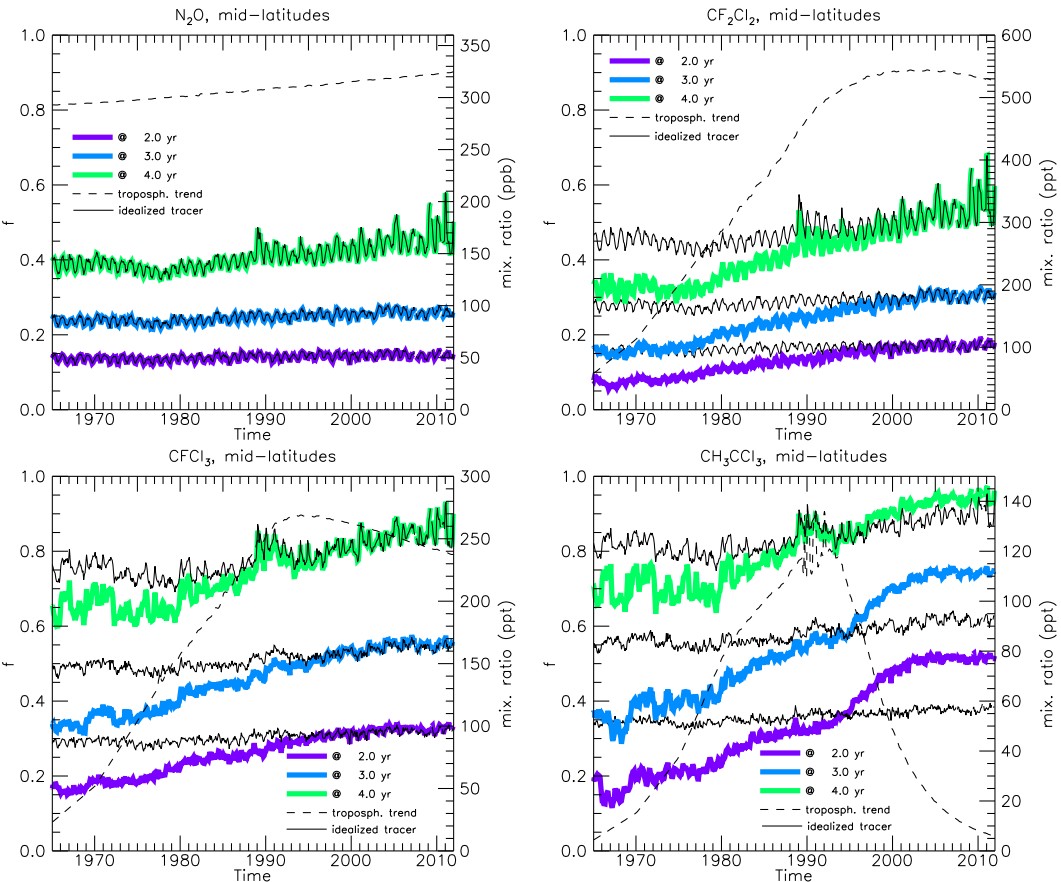

**Figure 3.** Temporal evolution of FRF calculated by the current formulation. Results for the realistic tracers are shown in color. The results for the idealized tracers (cf. Fig. 2) are shown as black lines for comparison. The related tropospheric trend of the species is plotted in dashed lines over the entire range in order to compare the magnitudes. There are obvious deviations between realistic and idealized tracers that depend on the tropospheric trends of the species (see text for explanation).

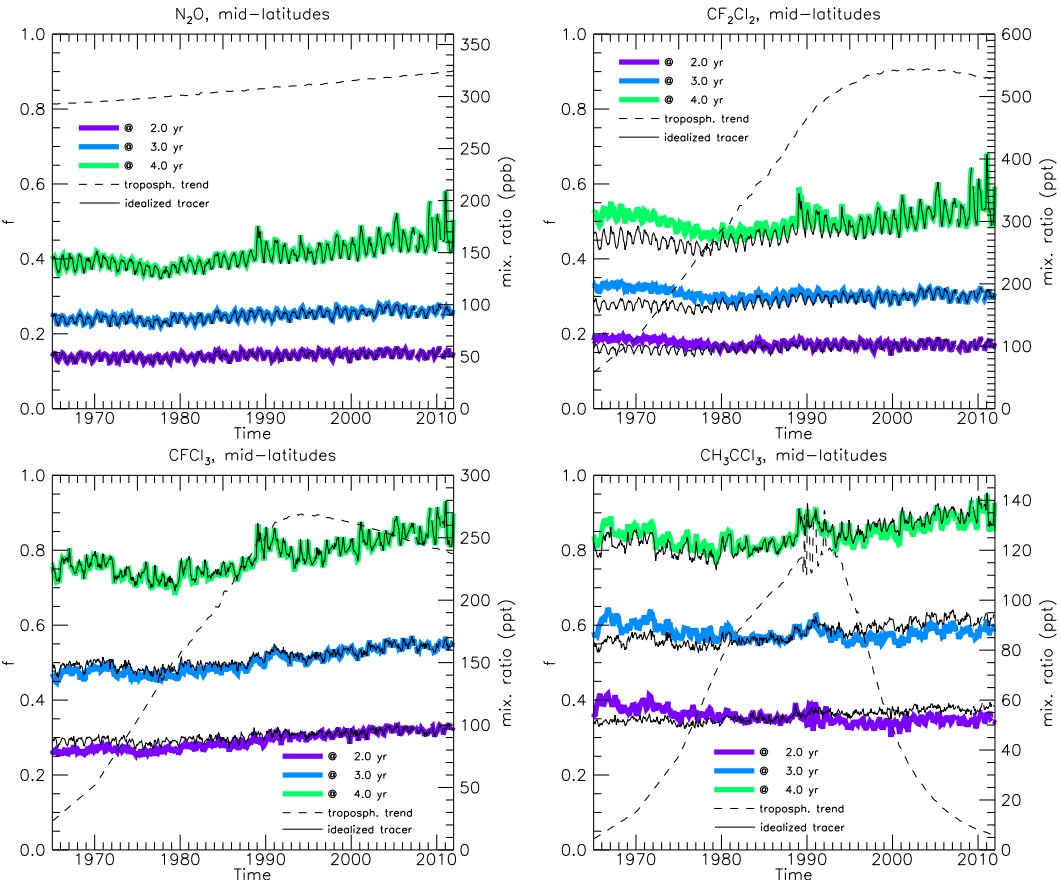

**Figure 4.** Temporal evolution of FRF calculated by the new formulation, taking into account chemical loss. The results of the realistic tracers are shown in color on different age isosurfaces. The results of the idealized tracers are shown in solid black lines whereas the tropospheric trend is plotted in dashed lines. We find a much better agreement between idealized and realistic tracers compared to the current formulation of FRF (cf. Fig. 3).