# Peer review of "A new time-independent formulation of fractional release"

_Atmospheric Chemistry and Physics, 2016_

## Referee Comment (RC1) · Anonymous Referee #1 · 29 Sep 2016

**General:**
A new definition of the fractional release factor (FRF) is proposed. It takes into account the fact that chemical loss during transport changes the weighting of different transit times. Consequently, a slightly changed definition of the reference tracer is proposed to quantify chemical loss in the stratosphere. The biggest advantage of the new FRF is that it removes some undesirable dependence on the trends of the considered species. This fact is validated with the model where halocarbons with idealized sources without trends as well as with realistic sources and trends are implemented. This important contribution is supported by well-performed figures. However, the way of explaining this approach has to be improved (see major point). Also the disadvantages of the new definition (the reference tracer is difficult to understand) are not discussed. I think that the very experienced co-authors could help to do this job. The paper may be

acceptable after a major revision improving these points.

**Major points:**

1. All definitions of the fractional release factors $f$ (FRF) have the form

$$f = \frac{R - \chi}{R} \qquad (1)$$

with $R$ being a kind of reference and $\chi$ denoting the mixing ratio of the considered species. All quantities are space-time functions, i.e. of $(r, t)$.

The first possible choice for the reference function $R$ is (your relations (1) and (2) following Salomon and Albritton, 1992):

$$R(r, t) = \chi_E(t - \Gamma(r, t)) \qquad (2)$$

with $\chi_E(t) = \chi(r = r_E, t)$ where $r_E$ means the entry point like the tropical tropopause. However, to simplify arguments I would recommend to use the Earth's surface where trends of all relevant species are known. The second possible choice according to Newman et al., 2007 is (your eq. (8)):

$$R(r, t) = \int_0^\infty \chi_E(r, t - \tau) G(r, t, t - \tau) \, d\tau . \qquad (3)$$

Note that (3) reduces to (2) only for a linear tracer (Hall 1994). Furthermore, definition (2) is very simple to use because only the knowledge of the mean age $\Gamma$ is necessary.

The third choice is your relation (19), i.e.:

$$R(r, t) = \int_0^\infty \chi_E(t - \tau) G_N^*(r, t, t - \tau) \, d\tau . \qquad (4)$$

Even if you get much better agreement with your idealized species without any trends, the interpretation of (4) is much less clear than for (2) and (3). The reference functions (2) and (3) can be understood as passively transported "reference species". This type of interpretation is much more difficult for definition (4).

My main criticism is that it is very difficult to get a clear picture what you did. I would recommend to rewrite both introduction and section 2. Even if I am familiar with the concept of age spectrum it was difficult for me to follow your arguments (and your notation).

**Minor points:**

1. Abstract and Introduction
   "steady state" - This is one of the central concepts related to the FRFs. FRFs should be independent of time even if the emissions have a trend and the dynamics (Brewer-Dobson circulation) is changing. You should more carefully introduce this concept. E.g. it is important to explain that the effect of changing dynamics cannot be removed and the effect of changing emissions should be removed and why it is so difficult to do it. All the definitions of $R$ as a ratio remove the dependence on emissions, or not ?. If not why ?

2. Abstract (L. 5)
   "for a given atmospheric situation" - do not understand what you want to say in context of "steady state"

3. Abstract (L. 8)
   "current formulation" - it is difficult to understand without reading the paper what you mean here

4. Abstract (L. 8)
   "tropospheric trends" - difficult to understand, maybe "trend in the emissions"

5. Abstract (L. 10)
   "reduces the time-dependence in correlations" - even after reading the paper I do not understand what you mean here. For me "reduces the time-dependence in FRFs" would be enough.

6. Introduction (P. 1, L. 20)
   ...which intensify ozone destruction. (sounds better for me)

7. Introduction (P. 2, L. 3)
   ...when the ODS are completely depleted.

8. Introduction (P. 2, L 14)
   "tropospheric trends": I would recommend to use "trends in the emissions" instead of "tropospheric trends" that does not sound very clear to me. Here is also the place where the concept of "steady state" should be explained in all details (see my first minor comment)

9. Introduction (P. 2, L 18)
   "current formulation" - please explain it more detailed (widely used method of calculation, especially for the calculation of ODS (citations...)

10. Introduction (P. 2, L 32)
    It is purpose of this paper to find a steady state formulation... - here you can see, how important is it to understand first why steady state formulation is so important

11. Section 2
    I would recommend, to reformulate following my "major point"

12. Section 3
    Starting from here, paper really improves

13. General question

Following your procedure, you significantly reduce the time dependence of the FRF calculated with realistic emissions. However, it seems to be like a "magic effect" without any deeper explanation why it works. Including chemistry into the age spectrum calculation is a formal step, but once again, all definitions of FRFs which are relative definitions (see my major point) should be independent on the trends of emissions.

---

## Referee Comment (RC2) · Anonymous Referee #2 · 18 Nov 2016

This is an interesting paper that promises an improved method to compute fraction release factors (FRF). It includes results that suggest great promise but as presently written it is does not make a convincing case. The paper is unclear and confusing in it derivations, which are at the heart of the paper. In addition, it makes exaggerated claims as to the expected properties of FRFs (e. g., their time-independence). Thus, I cannot recommend publication in ACP without major revisions.

Specific Comments

P3 L1-2: It is expected (even if undesired) that ODPs can change with time so not all time variation in the FRFs is incorrect. So a bit more discussion of this point is needed.

P4 L3: Do you mean Newman et al. (2007) here?

[Figure]

P4 eq (5): Should be $G(r, t')$ here.

P4 L11-23: I find this "derivation" sloppy, misleading, and possibly incorrect. The $f(t)$ defined in eq (1) is a function of time (i.e., model time). The $f(t')$ in eq (6) is a function of transit time. These are completely different and cannot be replaced by each other. You should write the $f$ in eq (6) as $f(r, t, t')$ and it would be better to use a new function name as well. $f^*$ may fit in well with your later discussion.

However, I strongly feel that L11-19 should just be removed from this section and fleshed out in more detail in section 5. Eq (8) is trivial to derive from eq (1) and (5) and without more discussion your alternative derivation is just confuses the matter. For example, why would $f(r, t, t')$ be independent of transit time $t'$? Clearly it should be strongly dependent on transit time. But you confuse the matter by saying just "time" on L18 and thus make it seem you are talking about model time, which are completely unrelated.

P4 eq (8): Should be $f(r, t)$ on the left side of equation.

P5 L2-3: You changed the meaning of the sentence in the Laube et al. paper. The full sentence is "Experimentally derived correlations of FRFs with mean age of air should thus only be considered as time-independent as long as there are no major changes in stratospheric transport or relative tropospheric growth rates." Seems reasonable enough. Not being time-independent is expected (e.g., from circulation changes) and does not render the concept invalid. This is not to say that all sources of time-dependence are ok, but just that a more nuanced discussion is needed.

P5 L6-7: There is no such inherent assumption. One can derive eq (8) from eq (1) and (5).

P5 L9-10: Again, this sentence is too absolute. Eq (8) is not expected to produce time-independent values (as stated many times already). The approach of eq (8) can only hope to remove time-dependence from source gas trends (which of course it may or

may not).

P6 L9: You do not state how you compute the age of air. Please discuss.

P6 L32 Mention you are starting to discuss figure 2. Took me a bit to figure out the change and figure 2 is never mentioned in the text.

P7 L6-17: This discussion needs to be moved up to the beginning of the section or perhaps to Section 3 since it affects all of the calculations shown in this section.

Section 4: I'd be interested in seeing the difference between eq (1) and (8), where in eq (1) a reasonable lag is used to account for the transit time. I get the feeling looking at your results that they may not be much worse using (1). I think this comparison should be added to the paper (it could be in a supplement).

Section 5: This derivation seems awfully convoluted. Certainly it could be cleaned up. And it will help if you move and expand the discussion around eq (6) and (7) to here, as discussed above. And please explicitly state that $f$ is a function of both $t$ and $t'$, $f(r, t, t')$, so it is more clear what is going on.

P10 L1: $\bar{f}$ does not depend on $t'$ but it is not a constant.

P10 L21-23: Now I'm confused. If you use the same function for $G_N^*$ as $G$, then eq (8) and eq (19) are identical. Thus, you should reproduce the results in figure 3 in figure 4. What have you gained in the derivation?

P11 L4-5: How are you using $\Gamma^*$? This is not explained at all. Perhaps this is at the heart of my confusion on the last point. I'm suspecting that your definition of age differs in figure 3 and 4. More description is greatly needed.

P11 L27-29: Seems like these details should have been mentioned in Section 3.

---

## Author Comment (AC1) · 13 Jan 2017

**Answers to the Referees**

January 13, 2017

We would like to thank both reviewers for their comments on the paper. We have largely followed the suggestion of the reviewers, which has resulted in major changes of the manuscript. These changes are detailed below. Regarding the structure of the manuscript, the major changes are that the discussion on what should lead to changes in FRF (changes in stratospheric transport or photochemistry) and what should not lead to changes in FRF (trends in tropospheric mixing ratios) has been detailed and that the last part of the former Sect. 2 has partly been included in the introduction and partly been deleted (as it was confusing). Also the discussion of how a chemically active tracer propagates into the stratosphere has now been concentrated in Sect. 5 (and been removed from Sect. 2).

In addition to the changes motivated by the comments of the reviewers, we have implemented a few changes which occurred during the revision. The major changes directly due to reviewer comments are as follows:

- In the introduction we revised the paragraph on mean age of air in the following way: *"In brief this concept of mean age of the air (Hall and Plumb, 1994; Waugh and Hall, 2002) relies on the idea that different transport pathways (and associated transit times) contribute to the chemical composition of an air parcel at a given point in the stratosphere. Different transit times associated to these different transport pathways have different probabilities, which are described by a probability density function (pdf) also known as age spectrum. By folding the probability distribution for a certain entry time into the stratosphere with the time series of the inert trace gas, its mixing ratio at this point in the stratosphere can be derived, as long as there is no chemical loss. The transit time distribution is called the age spectrum $G$ and the first moment (the arithmetic mean) is called mean age $\Gamma$. Plumb et al. (1999) showed that this concept is only valid to describe the propagation of inert tracers into the stratosphere. The underlying reason is that air parcels which have already spent a lot of time in the stratosphere, will only contribute very little to the observed mixing ratio of a compound which experiences photochemical loss, as a large fraction of the molecules of this compound will not be in the organic form anymore. Air parcels with long transit times thus need to be weighted less heavily then air parcels with short transit times."*

- In Sect. 2 we give a more detailed explanation of the entry mixing ratio and describe that it includes transport but not the chemical depletion on the transport pathway (a representation of the entry mixing ratio including chemical loss will then be derived in Sect. 5): *"In this representation of the entry mixing ratio the transport of the species to a certain location in the stratosphere is represented by $G$. It takes into account that several transit times and pathways are possible which is an improvement compared to the representation of the entry mixing ratio according to Eq. (2) where only a single transit time is allowed for. Nevertheless, Eq. (5) is only valid for chemically inert species and does not take respect to chemical processes."*

- In the discussion (Sect. 7) we added a paragraph on the discrepancies of FRFs deduced from observations: *"This [strong tropospheric trends] may lead to discrepancies in fractional release factors derived during different time periods. Such differences in FRF have been observed between the work of Laube et al. (2013) and Newman et al. (2007). The FRF values derived by Laube et al. (2013) were lower than those derived by Newman et al. (2007) on the 3 year mean age isosurface. As the tropospheric trends were lower during the observations used by Laube et al. (2013), it is expected that the re-calculation using our method should even increase the observed difference. We therefore conclude that the calculation of mean age may be the reason for the observed discrepancies, as suggested by Laube et al. (2013)."*

**Reply to Referee #1**

**General comment**

*A new definition of the fractional release factor (FRF) is proposed. It takes into account the fact that chemical loss during transport changes the weighting of different transit times. Consequently, a slightly changed definition of the reference tracer is proposed to quantify chemical loss in the stratosphere. The biggest advantage of the new FRF is that it removes some undesirable dependence on the trends of the considered species. This fact is validated with the model where halocarbons with idealized sources without trends as well as with realistic sources and trends are implemented. This important contribution is supported by well-performed figures. However, the way of explaining this approach has to be improved (see major point). Also the disadvantages of the new definition (the reference tracer is difficult to understand) are not discussed. I think that the very experienced co-authors could help to do this job. The paper may be acceptable after a major revision improving these points.*

We thank Referee #1 for the helpful comments, suggestions and the summary of the manuscript. We agree that the Section on the calculation methods of FRF can be improved and that it might be complicated to understand the physical meaning of our new formulation. Therefore we revised the introduction, Sect. 2 as well as Sect. 5 to give a better representation of the formalism. We also enlarged the discussion section where we respond to the disadvantages of the new definition. A detailed description of the changes in the manuscript is given according to the major and minor points (see below).

**Major points**

All definitions of the fractional release factors $f$ (FRF) have the form

$$f = \frac{R - \chi}{R} \tag{0.1}$$

with $R$ being a kind of reference and $\chi$ denoting the mixing ratio of the considered species. All quantities are space-time functions, i.e. of $(r, t)$.

The first possible choice for the reference function $R$ is (your relations (1) and (2) following Salomon and Albritton, 1992):

$$R(r, t) = \chi_E(t - \Gamma(r, t)) \tag{0.2}$$

with $\chi_E(t) = \chi(r = r_E, t)$ where $r_E$ means the entry point like the tropical tropopause. However, to simplify arguments I would recommend to use the Earth's surface where trends of all relevant species are known. The second possible choice according to Newman et al., 2007 is (your eq. (8)):

$$R(r, t) = \int_0^\infty \chi_E(r, t - \tau) G(r, t, t - \tau) d\tau. \tag{0.3}$$

Note that (3) reduces to (2) only for a linear tracer (Hall 1994). Furthermore, definition (2) is very simple to use because only the knowledge of the mean age $\Gamma$ is necessary.
The third choice is your relation (19), i.e.:

$$R(r, t) = \int_0^\infty \chi_E(t - \tau) G_N^*(r, t, t - \tau) d\tau. \tag{0.4}$$

Even if you get much better agreement with your idealized species without any trends, the interpretation of (4) is much less clear than for (2) and (3). The reference functions (2) and (3) can be understood as passively transported "reference species". This type of interpretation is much more difficult for definition (4).
My main criticism is that it is very difficult to get a clear picture what you did. I would recommend to rewrite both introduction and section 2. Even if I am familiar with the concept of age spectrum it was difficult for me to follow your arguments (and your notation).

Thank you for this nice overview of the formalism. It depends on the choice of the entry mixing ratio $\chi_{entry}$ (your reference function $R$) how the formulation of FRF changes. Thus it is the gist of this paper to find a proper representation of $\chi_{entry}$. We decided to keep to the term "entry mixing ratio" because we think that it is more striking. We agree that the physical meaning of Eq. (0.4) is less

intuitive than for Eq. (0.2) and (0.3). Therefore we expanded the discussion and interpretation of Eq. (0.4) in Sect. 5 (please also see the answer on your minor point No. 13).

Furthermore we revised the introduction of the article and expanded the description of steady state values and ODP therein. The discussion on inconsistencies of observed FRFs was moved from Sect. 2 to the introduction as we think it fits much better in this context. In addition, we explained the problems in the currently used formulation (Newman et al., 2007) in more detail which originates in the fact that the propagation of a chemically active species is treated as for an inert species.

We also reformulated Sect. 2 according to the recommendations of the referees. The discussion on the calculation of the stratospheric and the entry mixing ratio is now moved to Sect. 5 (also see comment 13 of Referee #2). In Sect. 2 we now only discuss the classical and the current formulation of FRF and state that it includes transport but no chemical loss processes which is inappropriate for chemically active species. Based on that issue we start/construct the derivation of the new formulation in Sect. 5.

Sect. 6 was complemented by a more detailed description of the applied arrival time distribution $G_N^*$ which is an inverse Gaussian distribution with the parameters $\Gamma^*$ and $\Delta^*$. *"We choose $G_N^*$ to have the same shape as G, i.e. an inverse Gaussian distribution but with the parameters $\Gamma^*$ (first moment) and $\Delta^*$ (second moment), so that $G_N^* = G(\Gamma^*, \Delta^*, t')$. Like for G we use a constant ratio of the squared width to mean age of $\Delta^{*2}/\Gamma^* = 0.7$ yr."*

In the discussion (Sect. 7) we added a paragraph on the disadvantages of the new formulation: *"We also acknowledge that the new formulation is less intuitive than the formulation used by Newman et al. (2007) and Laube et al. (2013). However, as we have shown that the method used by Laube et al. (2013) and Newman et al. (2007) yields values which are strongly influenced by the tropospheric trend, this loss of intuitivity and the added dependence on model information is necessary, as much more representative values are derived."*

In addition, we also give a short sensitivity study on the importance of the lifetime that needs to be assumed in the calculation of the mean arrival time $\Gamma^*$. *"The parameterisation given by Plumb et al. (1999) depends on the stratospheric lifetime of the species. As fractional release also depends on the lifetime, one may argue that there is a certain circular argumentation involved. Indeed, if the assumption on stratospheric lifetime is very far off, and tropospheric trends are large then our new method will also fail in correcting for the tropospheric trend. However, it should be noted that the calculation is not extremely sensitive to the assumed lifetime. We investigated the sensitivity for a CFC-12 like tracer with a linearly increasing trend of 5 % per year. For an assumed steady state FRF of 0.5 at a mean age of 4 years using our method, a value of 0.5 is found with a deviation of 0.5 % for an uncertainty in the assumed lifetime of 20 %. Using the current method ignoring the effect of chemical loss would result in an FRF of 0.45, i.e. 10 % lower than the correct value. The sensitivity to the assumed lifetime is thus rather small."*

**Minor points**

1. *Abstract and Introduction*

   *"steady state" - This is one of the central concepts related to the FRFs. FRFs should be independent of time even if the emissions have a trend and the dynamics (Brewer-Dobson circulation) is changing. You should more carefully introduce this concept. E.g. it is important to explain that the effect of changing dynamics cannot be removed and the effect of changing emissions should be removed and why it is so difficult to do it. All the definitions of R as a ratio remove the dependence on emissions, or not ?. If not why ?*

   FRFs should be independent of tropospheric trends but not of changes in the stratospheric conditions. This means that FRF is expected to change if e.g dynamics or the photochemistry changes. The purpose of our article is to find a new formulation of FRF that does not depend on tropospheric trends. Our formulation cannot (and does not want to) remove changes due to modified stratospheric dynamics. We go into details on the discussion about steady-state quantities in the introduction where we explain why FRF should be a time-independent quantity. In the revised text we added *"FRF should thus be specific for a given molecule and a given atmospheric condition. If atmospheric conditions, e.g. stratospheric dynamics or the actinic flux responsible for photochemical degradation, change, FRF is expected to change. However, FRF should not be dependent on the tropospheric trend of the chemical compound under otherwise unchanged atmospheric conditions."*

   In Sect. 2 we discuss the influence of tropospheric trends on the calculation of the entry mixing

ratio (your reference function $R$) which is the crucial variable in the calculation of FRF. In Sect. 2 we thus added *"In case of a chemical compound which is in steady state between emissions into the atmosphere and atmospheric loss the tropospheric trend will be zero and $\chi_{entry}$ will just be its tropospheric mixing ratio. However, if the tropospheric mixing ratio of the trace gas changes with time, $\chi_{entry}$ must be calculated based on assumptions about stratospheric transport. As most ozone depleting substances are not in steady state but have tropospheric trends, this needs to be taken into account in calculating the entry mixing ratio $\chi_{entry}$. It is through the calculation of $\chi_{entry}$ that the time independence of FRF should be achieved."*

2. *Abstract (L. 5)*

   *"for a given atmospheric situation" - do not understand what you want to say in context of "steady state"*

   We moved the discussion on that into the introduction. The "atmospheric situtation" or "atmospheric condition" is the interplay between different atmospheric processes like dynamics, radiation and photochemistry. Please also see the comment to point No. 1.

3. *Abstract (L. 8)*

   *"current formulation" - it is difficult to understand without reading the paper what you mean here*

   The term was changed to "widely used formulation" in the abstract as to avoid citations therein. In the introduction and Sect. 2 we explain that the "current formulation" is the calculation method of FRF given by Newman et al. (2007) and applied e.g. by Laube et al. (2010,2013). After introducing the relevant equation we stated: *"Subsequently we will refer to Eq. (6) as the "current formulation of FRF" as it has been used in Newman et al. (2007); Laube et al. (2013)."*

4. *Abstract (L. 8)*

   *"tropospheric trends" - difficult to understand, maybe "trend in the emissions"*

   Using the term "tropospheric trends" we mean the trends in the tropospheric mixing ratios and not in the emissions. Thus we changed the term to *"trends in the tropospheric mixing ratios"* in the abstract.

   There is a difference between the trend in the emissions and in the mixing ratio: the tropospheric mixing ratios are influenced by the balance between the emissions and loss processes. A reduction of the emissions (e.g. due to the Monteal Protocol) does not instantaneously lead to a reduction in the mixing ratios when the removal processes are unchanged. Thus the trend in the emissions does not coincide with the trend in the mixing ratios which we denote as the "tropospheric trend".

5. *Abstract (L. 10)*

   *"reduces the time-dependence in correlations" - even after reading the paper I do not understand what you mean here. For me "reduces the time-dependence in FRFs" would be enough.*

   The "time-dependence in correlations" relates to the work of Plumb et al. (1999) who developed a method to reduce the time-dependence in the correlations between two tracers with stratospheric loss and strongly changing tropospheric abundances. The reference (Plumb et al., 1999) was removed from the abstract of a former version of the manuscript. We agree that this might be confusing and changed the sentence according to the referees suggestion. *"Taking into account chemical loss in the calculation of stratospheric mixing ratios reduces the time-dependence in FRFs."*

6. *Introduction (P. 1, L. 20)*

   *...which intensify ozone destruction. (sounds better for me)*

   We agree to the reviewer, changed the sentence and also added a reference: *"The gases are emitted in the troposphere, where many of them are nearly inert before they enter the stratosphere at the tropical tropopause. In the stratosphere, many of the molecules will be broken down photochemically and release halogen radicals that intensify ozone destruction (Solomon, 1990)."*

7. *Introduction (P. 2, L. 3)*

   *...when the ODS are completely depleted.*

We adopt the suggestion of the Referee and completed the sentence in the following way *"The FRF increases until it reaches the value of 1 when the ODS is completely depleted and all halogen atoms it contained have been released."*

8. *Introduction (P. 2, L 14) "tropospheric trends": I would recommend to use "trends in the emissions" instead of "tropospheric trends" that does not sound very clear to me. Here is also the place where the concept of "steady state" should be explained in all details (see my first minor comment)*

There is a difference in the "trend in the emissions" and "the tropospheric trend" used here. What we used is the trend in the tropospheric mixing ratios. In case of a steady state between the emissions and atmospheric loss, the tropospheric mixing ratio corresponds to the entry mixing ratio needed to calculate FRF according to Eq. (0.1). Otherwise, the entry mixing ratio needs to be calculated from the convolution of the tropospheric time series and the transit time distribution. Please also see the answer on comment No. 4.

We added a more distinct explanation of what we mean by the term "tropospheric trend" in the introduction: *"However, FRF should not be dependent on the trend in the tropospheric mixing ratios of the chemical compound (tropospheric trend) under otherwise unchanged atmospheric conditions."*

9. *Introduction (P. 2, L 18)*

*"current formulation" - please explain it more detailed (widely used method of calculation, especially for the calculation of ODS (citations...)*

We added a the citation of Newman et al. (2007) after the first use of the term "current formulation" in the introduction of the manuscript. In Sect. 2 we then introduce the different formulations of FRF which differ in the calculation of the entry mixing ratio $\chi_{entry}$. In this section we present the formulations given by Solomon and Albritton (1992) ("first formulation") as well as Newman et al. (2007) which we call the "current formulation" as it is the most recent definition of FRF. Please also see comment on point No. 3.

10. *Introduction (P. 2, L 32)*

*It is purpose of this paper to find a steady state formulation... - here you can see, how important is it to understand first why steady state formulation is so important*

We deleted this particular sentence in the revised version but deepened the discussion on steady state in the introduction as explained in the comments on point 1 and 2.

11. *Section 2*

*I would recommend, to reformulate following my "major point"*

We revised Sect. 2 where we now only present the general definition of FRF and the formulation of the entry mixing ratio given by Solomon and Albritton (1992) (Eq. (0.2)) as well as by Newman et al. (2007)(Eq. (0.3)). We removed the formalism which shows the problems of the currently used formulation from Sect. 2 as it seems to be more suitable in Sect. 5, which was also suggested by Referee #2. The derivation of the new entry mixing ratio (Eq. (0.4)) remained in Sect. 5 as it is part of our new formulation of FRF.

12. *Section 3*

*Starting from here, paper really improves*

Thank you.

13. *General question*

*Following your procedure, you significantly reduce the time dependence of the FRF calculated with realistic emissions. However, it seems to be like a "magic effect" without any deeper explanation why it works. Including chemistry into the age spectrum calculation is a formal step, but once again, all definitions of FRFs which are relative definitions (see my major point) should be independent on the trends of emissions.*

The reviewer is correct, a detailed physical interpretation of our new formulation was missed in the manuscript. That might be the reason why the improvement of the new formulation and the good agreement of the realistic and the idealized tracers seems surprising. The reason for the

deficient results of the current formulation is that the propagation of a chemically active tracer is treated as for an inert tracer.

Therefor we explicitly added a discussion on the new entry mixing ratio in Sect. 5. In the revised version, we added the following explanation: *"Using $G_N^*$ instead of $G$ results in a lesser weighting of the tail of the transit time distribution which is reasonable, especially for CAS with short lifetimes. A shorter-lived species is almost completely depleted after a transit time of e.g. 4 years thus this transit time $t'$ should not contribute in the convolution with the tropospheric time series when calculating the remaining organic fraction. For such shorter lived species the remaining amount in the original organic form is thus hardly influenced by the tropospheric mixing ratios of air which entered a long time ago (the "tail" of the age spectrum for an inert trace gas). The shorter lived the trace gas is, the more the weighting needs to be shifted to the short fraction of the age spectrum. The arrival time distribution describes the relevant weighting of the different transit times and is specific for each trace gas."*

**Reply to Referee #2**

**General comment**

*This is an interesting paper that promises an improved method to compute fraction release factors (FRF). It includes results that suggest great promise but as presently written it is does not make a convincing case. The paper is unclear and confusing in it derivations, which are at the heart of the paper. In addition, it makes exaggerated claims as to the expected properties of FRFs (e. g., their time-independence). Thus, I cannot recommend publication in ACP without major revisions.*

Both reviewers seem to have had problems to understand our argumentation. We believe that the main problem is the presentation, as none of the reviewers has identified a mistake in our new formulation (see also answer to point 4 of Referee #2). We have significantly changed the presentation according to the suggestions of the reviewers and believe that the presentation is now much clearer.

**Specific Comments**

1. *P3 L1-2:*

   *It is expected (even if undesired) that ODPs can change with time so not all time variation in the FRFs is incorrect. So a bit more discussion of this point is needed.*

   In the revised introduction and Sect. 2 we put emphasis on the differentiation between temporal changes in FRFs (and ODPs, respectively) due to tropospheric trends and due to changes in the dynamics or chemistry. If e.g. the stratospheric circulation changes, the FRFs are also expected to change. But the calculated FRFs should be independent of tropospheric trends. We explained this issue in the following paragraph: *"FRF thus describes the effectiveness with which a certain ODS is broken down in the stratosphere. For the same time spent in the stratosphere, shorter lived species will have higher FRF than longer lived molecules. FRF are therefore used in the calculation of the Ozone Depletion Potential (ODP), a quantity which describes how effective a certain chemical is at destroying stratospheric ozone (Solomon et al., 1992). FRF should thus be specific for a given molecule and a given atmospheric condition. If atmospheric conditions, e.g. stratospheric dynamics or the actinic flux responsible for photochemical degradation, change, FRF is expected to change. However, FRF should not be dependent on the trend in the tropospheric mixing ratios of the chemical compound (tropospheric trend) under otherwise unchanged atmospheric conditions."*

   In Sect. 4 we also added a paragraph on the discussion of Fig. 3. We discuss the difference between the changes of FRF due to changing dynamics and chemistry and the change due to tropospheric trends. Because the idealized tracer is free from tropospheric trends, the change must be caused by changes in dynamics or chemistry. The change of FRF derived from the realistic tracers is influenced both by circulation changes as well as changes due to tropospheric trends. This variation is about one order of magnitude bigger than the variation solely caused by changes in circulation. *"It is obvious from Fig. 3 that the changes in FRF calculated for the realistic tracers are much larger than for the idealized tracers. The variation in the idealized tracers reflects the changes due to changing chemistry and dynamics. As the only difference between the idealized and the realistic tracers is the tropospheric trend of the realistic tracers, the larger variability of FRF for the realistic tracers must be due to the way that the tropospheric trend is considered in the calculation of FRF according to the current formulation."*

2. *P4 L3:*

   *Do you mean Newman et al. (2007) here?*

   Thank you, we changed the reference to Newman et al. (2007).

3. *P4 eq (5):*

   *Should be $G(r, t_0)$ here.*

   Done. In our notation with transit time $t'$: $G(r, t')$

4. *P4 L11-23:*

   *I find this "derivation" sloppy, misleading, and possibly incorrect. The $f(t)$ defined in eq (1) is a function of time (i.e., model time). The $f(t_0)$ in eq (6) is a function of transit time. These*

*are completely different and cannot be replaced by each other. You should write the f in eq (6) as*
*$f(r, t, t_0)$ and it would be better to use a new function name as well. $f^*$ may fit in well with your*
*later discussion.*
*However, I strongly feel that L11-19 should just be removed from this section and fleshed out in*
*more detail in section 5. Eq (8) is trivial to derive from eq (1) and (5) and without more discussion*
*your alternative derivation is just confuses the matter. For example, why would $f(r, t, t_0)$ be*
*independent of transit time $t_0$? Clearly it should be strongly dependent on transit time. But you*
*confuse the matter by saying just "time" on L18 and thus make it seem you are talking about*
*model time, which are completely unrelated.*

While we admit that the explanation of the derivation of the new formulation of FRF may have
been poorly explained, we do not agree that it is sloppy or even could be possibly wrong. In
particular this latter point (possibly wrong) is not explained by the reviewer. We suspect that
this perception may have been caused by the way we presented the new formulation. This has
been improved, as also explained in the answers to Referee #1.

We agree to the referee that the discussion on Eq. (6) should be part of Sect. 5 rather than
Sect. 2. Thus we removed the equations on the stratospheric mixing ratio $\chi_{strat}$ (Eq. (6) and
(7)) from Sect. 2 and included them into the derivation of the new formulation (Sect. 5). In
the revised version of Sect. 2 we now only present the general formulation of FRF (Eq. (1)) and
show two possible choices for the entry mixing ratio $\chi_{entry}$ according to Solomon and Albritton
(1992) and Newman et al. (2007). On this basis we deduce the "current formulation" of FRF (Eq.
(8)). As FRF should not depend on tropospheric trends, we make the assumption that $f$ does not
depend on time $t$ (thus $f = f(r)$), as long as there are no changes in stratospheric transport (also
see our answer to point No.1). We expressed this as follows: *"As explained above, FRF should*
*be independent of the tropospheric trend of the species, but is expected to change if atmospheric*
*conditions, especially stratospheric dynamics or photochemistry change. For the discussion of the*
*different methods of calculating FRF, we assume that stratospheric transport is stationary in time,*
*i.e. that the average circulation does not change with time. Under this assumption of unchanged*
*stratospheric transport, the fractional release factor $f(r, t)$ should not change with time, and thus*
*be $f(r)$, independent of $t$."*

In Sect. 2 we also expanded the description of transit times and the distinction between $t$ (time)
and $t'$ (transit time): *"Note that in the following we will use $t$ to denote time, whereas transit*
*time (i.e. the time of a fluid element spent in the stratosphere) is denoted as $t'$. The distribution*
*of the probabilities of the different transit times is called the age spectrum. It is denoted $G$.*
*Assuming that the average stratospheric transport is stationary in time (i.e. no long term changes*
*in stratospheric dynamics) the probability for a certain transit time $t'$ will only be a function of*
*the location in the stratosphere, thus $G = G(r, t')$. In particular the age spectrum will then only*
*depend on the location $r$ in the stratosphere and is not a function of time $t$. For simplicity, we will*
*make this assumption of unchanged dynamics in the following, as FRF at a given location should*
*be unchanged as long as the stratospheric transport is unchanged."*

In Sect. 5 we show the problems/ controversy in the current formulation of FRF pointing out
that $f$ is a function of the location $r$ as well as transit time $t'$ (which is ignored in the current
formulation of FRF). This is reasonable as air parcels with long transit times certainly suffered a
stronger chemical degradation than air parcels with short transit times. We added: *"This fraction*
*$f$ will be a function of the transit pathway and the transit time $t'$. For simplification, we assume*
*that longer transit pathways will be linked with more chemical loss and longer transit times, thus*
*we consider $f$ to be a function of $t'$ and location $r$ only."*

As $f$ is a function of transit time $t'$ it cannot be extracted from the integral in Eq. (6) (stratospheric
mixing ratio). On this basis we develop the new definition of FRF by introducing the new arrival
time distribution $G^*$ which includes chemical loss (by $f = f(r, t')$). We decided to keep to the
denotation of $\bar{f}$ as it represents an arithmetic mean fractional release value.

A discussion on the new entry mixing ratio follows *"The entry mixing ratio in this new formulation*
*now takes into account transport as well as chemical loss processes. Using $G_N^*$ instead of $G$ results*
*in a lesser weighting of the tail of the transit time distribution which is reasonable, especially for*
*CAS with short lifetimes. A short-lived species is almost completely depleted after a transit time of*
*e.g. 4 years thus this transit time $t'$ should not contribute in the convolution with the tropospheric*
*time series. For such shorter lived species the remaining amount in the original organic form is*

*thus hardly influenced by the tropospheric mixing ratios of air which entered a long time ago (the "tail" of the age spectrum for an inert trace gas). The shorter lived the trace gas is, the more the weighting needs to be shifted to the short fraction of the age spectrum. The arrival time distribution describes the relevant weighting of the different transit times and is specific for each trace gas."*

5. *P4 eq (8):*

   *Should be $f(r,t)$ on the left side of equation.*

   FRF should be independent of time $t$ but it depends on transit time $t'$. Thus it should be $f(r,t')$ (please also see the reply to your point No. 4).

6. *P5 L2-3:*

   *You changed the meaning of the sentence in the Laube et al. paper. The full sentence is "Experimentally derived correlations of FRFs with mean age of air should thus only be considered as time-independent as long as there are no major changes in stratospheric transport or relative tropospheric growth rates." Seems reasonable enough. Not being time-independent is expected (e.g., from circulation changes) and does not render the concept invalid. This is not to say that all sources of time-dependence are ok, but just that a more nuanced discussion is needed.*

   FRFs are expected to change if e.g. stratospheric transport changes but it should not change due to changes in the tropospheric growth rates. We removed the citation of Laube et al. (2013) from the manuscript and also moved the discussion on the discrepancies in the observations from Sect. 2 into the introduction. Therein we now discuss possible reasons for the differences in the observations. It is likely that the differences originate in an insufficient correction of the tropospheric trends as observations took place in times when the trends in the tropospheric mixing ratios were very strong. *"In recent years, inconsistencies between FRF values derived from independent observations at different times were identified (Laube et al, 2013; Carpenter et al., 2014). This could either be caused by real changes in FRF, due to changing atmospheric conditions, or by deficiencies in the way that the tropospheric trends are taken into account in the calculation of FRF. The latter is very likely, as data from different time periods are compared, where trends differ not only in magnitude but sometimes even in the direction (positive/ negative trend), suggesting possibly large impacts of the way that tropospheric trends are considered in the calculation of FRF."*

7. *P5 L6-7:*

   *There is no such inherent assumption. One can derive eq (8) from eq (1) and (5).*

   We wanted to show that it is vice versa: Eq. (1) can be derived from Eq. (6) under the assumption that $f$ is independent of transit time $t'$. This discussion is now referred to Sect. 5 (also see our reply to point No. 4).

8. *P5 L9-10:*

   *Again, this sentence is too absolute. Eq (8) is not expected to produce time-independent values (as stated many times already). The approach of eq (8) can only hope to remove time-dependence from source gas trends (which of course it may or may not).*

   In the revision of Sect.2 this sentence was deleted, but the discussion on which changes should lead to changes in FRF has been expanded (also see reply to point No. 4).

9. *P6 L9:*

   *You do not state how you compute the age of air. Please discuss.*

   We added the description at the beginning of Sect. 4 where we discussed the correlations of FRF with age of air. *"In EMAC, the age of air is calculated from a diagnostic tracer. This tracer is nudged towards a linearly in time increasing mixing ratio at the lowest model layer."*

10. *P6 L32:*

    *Mention you are starting to discuss figure 2. Took me a bit to figure out the change and figure 2 is never mentioned in the text.*

    Thank you, we now refer to Fig. 2 in the text.

11. *P7 L6-17:*

   *This discussion needs to be moved up to the beginning of the section or perhaps to Section 3 since it affects all of the calculations shown in this section.*

   The reviewer is correct. In the beginning of Sect. 4 we discussed the correlations of FRF with mean age of air. For the calculation of FRF according to Eq. (8) it is necessary to know how the tropospheric time series $\chi_{trop}$ and the transit time distribution $G$ are computed and how the integral is being solved. We therefore moved the general description of the specific calculation method to the beginning of Sect. 4. We also added a paragraph on the matter that we did our analysis of the temporal evolution of FRF on age of air surfaces instead of space coordinates: *"Assuming that the age spectra for different locations with the same mean age are similar, we investigate changes of FRF in the model on age isosurfaces instead of on geographical coordinates. As mean age e.g. at a given location shows some variability with time, this is expected to lead to reduced variability."*

12. *Section 4:*

   *I'd be interested in seeing the difference between eq (1) and (8), where in eq (1) a reasonable lag is used to account for the transit time. I get the feeling looking at your results that they may not be much worse using (1). I think this comparison should be added to the paper (it could be in a supplement).*

   Eq. (1) is a general equation for the calculation of FRF. One needs to choose a representation of the entry mixing ratio to calculate FRF according to Eq. (1). Therefore Eq. (1) cannot simply be used to calculate FRF.

13. *Section 5:*

   *This derivation seems awfully convoluted. Certainly it could be cleaned up. And it will help if you move and expand the discussion around eq (6) and (7) to here, as discussed above. And please explicitly state that $f$ is a function of both $t$ and $t_0$, $f(r, t, t_0)$, so it is more clear what is going on.*

   We revised the structure of Sect. 5 according to the referee. We agree that the break between the discussion in Sect. 2 and Sect. 5 might be confusing and not helpful to understand the concept. The derivation of the new formulation of FRF now starts with a general consideration of how the stratospheric mixing ratio of a CAS can be calculated (this previously was part of Sect. 2): *"The mixing ratio of a chemically active substance at some point $r$ in the stratosphere at some time $t$, $\chi_{strat}(r, t)$, can be calculated by convoluting three functions: the tropospheric time series $\chi_{trop}(t - t')$, the remaining fraction due to photochemical loss $(1 - f(r, t'))$, and the transit time distribution or age spectrum $G(r, t')$, which is a function of transit time $t'$ and the location in the stratosphere $r$. As explained in Sect. 2, $G$ and $f$ are not functions of time $t$ as stratospheric transport is assumed to be stationary in time."*

$$\chi_{strat}(r, t) = \int_0^\infty \chi_{trop}(t - t')\,(1 - f(r, t'))\,G(r, t')dt'$$

   We also give an interpretation of the stratospheric mixing ratio: *"Physically Eq. (7) describes that the observed mixing ratio of a CAS will be the sum over the mixing ratios of the individual fluid elements with different transit times, different photochemical loss and different original mixing ratios upon entry into the stratosphere. For short lived species, the fluid elements with long transit times will contribute very little to the observed mixing ratio in the stratosphere, as the original content has been photochemically depleted. The tropospheric mixing ratio at that time is thus not very relevant for the observed mixing ratio. Imagine that a CAS has a decreasing trend in the troposphere and that its fractional loss will be nearly complete after a transit time of a 4 years. The observed mixing ratio on the 4 year iso agesurface will then be dominated by the short fraction of the transit time distribution, whereas longer transit times must be weighted less heavily. The probability density function describing how strongly which transit time and thus the corresponding tropospheric mixing ratio must be weighted should thus be different for species with different chemical loss and in particular also different for species with chemical loss then for species without chemical loss."*

Starting from here, we derive the new arrival time distribution $G^*$ (which includes chemical loss) and by normalizing $G^*$ we receive the expression

$$\bar{f}(r) \equiv \int_0^\infty f(r,t')G(r,t')dt' \quad \text{resp.} \quad \bar{f}(r) = \frac{\int_0^\infty \chi_{trop}(t-t')G_N^*(r,t')dt' - \chi_{strat}(r,t)}{\int_0^\infty \chi_{trop}(t-t')G_N^*(r,t')dt'}$$

which we interpret as the the mean fractional release: *"The new mean fractional release factor $\bar{f}$ should be independent of tropospheric trends and is only expected to change if stratospheric transport or photochemistry change."*

A discussion on the new entry mixing ratio follows. Please also see our reply to your point No. 4.

14. *P10 L1:*

   *$\bar{f}(r)$ does not depend on $t_0$ but it is not a constant.*

   This is correct, we changed the sentence to *"$\bar{f}$ does not depend on transit time $t'$ and can thus be extracted from the integral".*

15. *P10 L21-23:*

   *Now I'm confused. If you use the same function for $G_N^*$ as $G$, then eq (8) and eq (19) are identical. Thus, you should reproduce the results in figure 3 in figure 4. What have you gained in the derivation?*

   $G_N^*$ and $G$ have the same shape of an inverse Gaussian distribution with parameters $\Gamma$ (first moment, mean age) and $\Delta$ (second moment, width). But the functions $G_N^*$ and $G$ are not identical, they differ in their first moment and width. The first moment of the transit time distribution $G$ is the mean age $\Gamma$. The concept is only valid for chemically inert species. Plumb et al. (1999) derived a paramterization of a species dependent "mean arrival time" $\Gamma^*$ for chemically active species from a delta pulse emission calculation/ experiment. It takes into account that molecules with long arrival times travelled through regions with greater photochemical loss. In the beginning of Sect. 6 we added: *"We choose $G_N^*$ to have the same shape as $G$, i.e. an inverse Gaussian distribution but with the parameters $\Gamma^*$ (first moment) and $\Delta^*$ (second moment), so that $G_N^* = G(\Gamma^*, \Delta^*, t')$. Like for $G$ we use a constant ratio of the squared width to mean age of $\Delta^{*2}/\Gamma^* = 0.7$ yr. Plumb et al. (1999) derived a parameterization of a species dependent "mean arrival time" $\Gamma^*$ for a wide range of chemically active species from a delta pulse emission calculation. $\Gamma^*$ can be calculated from the mean age of air $\Gamma$ and the mean stratospheric lifetime $\tau$ by a parameterization scheme Plumb et al. (1999)."*

16. *P11 L4-5:*

   *How are you using $\Gamma^*$? This is not explained at all. Perhaps this is at the heart of my confusion on the last point. I'm suspecting that your definition of age differs in figure 3 and 4. More description is greatly needed.*

   This is correct, the definition of age differs in Fig. 3 and Fig. 4. In Fig. 3 we used the age of air $\Gamma$ in the calculation of the age spectrum $G = G(\Gamma)$ and with that in the calculation of the entry mixing ratio and FRF according to Eq. (5). In Fig. 4 instead, the FRF was calculated according to Eq. (19). The new arrival time distribution $G_N^* = G(\Gamma^*)$ is used in this calculation, where $\Gamma^*$ is the species dependent mean arrival time (Plumb et al., 1999). *"Using $\Gamma^*$ instead of $\Gamma$ takes respect to the chemical loss occurring on the transport pathways. We computed $\Gamma^*$ for the considered species and applied it as the first moment of our new arrival time distribution $G_N^*$."*

17. *P11 L27-29:*

   *Seems like these details should have been mentioned in Section 3.*

   We do not think that the description of the analyzed data (zonally averaged, monthly mean, latitude band) is part of the general model description (Sect. 3). We explained the considered model domain in Sect. 4, specifying the FRF in EMAC simulations. We think that a short description of the data should also be part of the summary.

On behalf of all authors,

Jennifer Ostermöller
January 13, 2017

[revised manuscript text omitted]

---

## Referee Report (RR1)

Review of the revised version of the paper:

"A new time-independent formulation of fractional release"

written by Jennifer Ostermller et al.

**General:**

The paper significantly improved. Especially, the introduction and the description of the problem are well-written and well-done. I have only few minor comments listed below.

**Minor points:**

1. Abstract, L3-5

   Here my recommendation: ...affects the calculation of the Ozone Depletion Potential (ODP). In this context time-independent values are needed which, in particular, should be independent of the trends in the tropospheric mixing ratios....

   I would not use the notation "steady-state" here because steady-state can also be understood in the context of steady state of forcing (i.e. trends of halogenated trace gases) or steady state of the whole atmosphere (no change in the BD circulation). What you mean is the independence of the definition of FRF on the tropospheric trends of the halogenated trace gases.

2. P2 L12

   "However, FRF should not be dependent..." - great sentence. I would like to have something similar in the abstract (see above)

3. P3 L31

   You use the notation "entry mixing ratio" at the location $r$ for $\chi_{entry}(r,t)$. For me it is much more a reference mixing ratio which has to be defined in appropriate way. Maybe you use this notation because of "historical reason". This could be OK if you explain it in this way. I would recommend to add 1-2 sentences to clarify this point.

4. P4 eq (2)

   I would recommend to write:

$$\chi_{entry}(r,t) = \chi_{trop}(t - \Gamma(r))$$

   where $\Gamma(r)$ is the mean age of a steady state atmosphere (see eq. (3)).

5. P5 L17-19

   I would remove these 2 sentences. They confuse more than explain

6. P6 L25

   Maybe more simple: "This tracer is linearly increasing in time in the lowest model layer"

7. P7 L25

   You should discuss different lifetimes of your idealized tracers earlier in the text, e.g. at the end of section 3.1 (after L21). It would make the following text easier to understand. Maybe you simply shift the lifetime discussion to this place.

---

## Author Response (AR2)

**Answers to Referee #1**
**(revised version of the paper)**

February 7, 2017

We would like to thank Referee #1 for the additional comments on the revised paper. We followed the suggestions of the reviewer and especially avoided the notation "steady state" in the abstract as well as in the further text. The changes according to the minor comments are detailed below.

**General comment**

*The paper significantly improved. Especially, the introduction and the description of the problem are well-written and well-done. I have only few minor comments listed below.*

Thank you.

**Minor points**

1. *Abstract, L3-5*

   *Here my recommendation: ...affects the calculation of the Ozone Depletion Potential (ODP). In this context time-independent values are needed which, in particular, should be independent of the trends in the tropospheric mixing ratios....*

   The referee is correct. We want to present FRF values that are independent of tropospheric trends and not a steady state formulation. We admit that the notation "steady state" is inappropriate in this context and rephrased the sentence according to the referee. *"In this context time-independent values are needed which, in particular, should be independent of the trends in the tropospheric mixing ratios (tropospheric trends) of the respective halogenated trace gases. For a given atmospheric situation, such FRF values would represent a molecular property."*

2. *P2 L12*

   *"However, FRF should not be dependent..." - great sentence. I would like to have something similar in the abstract (see above)*

   Please see our answer on comment No. 1.

3. *P3 L31*

   *You use the notation "entry mixing ratio" at the location $r$ for $\chi_{entry}(r,t)$. For me it is much more a reference mixing ratio which has to be defined in appropriate way. Maybe you use this notation because of "historical reason". This could be OK if you explain it in this way. I would recommend to add 1-2 sentences to clarify this point.*

   Yes, the notation "entry mixing ratio" was chosen for historical reasons. E.g. Daniel et al. (1995) and Schauffler et al. (2003) used this notation for the reference mixing ratio. We added a sentence in the beginning of Sect. 2 to explain that we keep to this term. *"The representative average entry mixing ratio $\chi_{entry}$ should thus be derived in a way that $f$ is time-independent. To be consistent with previous work (Daniel et al., 1995; Schauffler et al., 2003), we will refer to this quantity as "entry mixing ratio" in the following."*

4. *P4 eq (2)*

   *I would recommend to write:*

   $$\chi_{entry}(r,t) = \chi_{trop}(t - \Gamma(r))$$

*where $\Gamma(r)$ is the mean age of a steady state atmosphere (see eq. (3)).*

We added the dependence of mean age from the location in the stratosphere in Eq. (2) but to avoid the notation of "steady state" (please also see comment No. 1), we decided to keep to our explanation of mean age.

5. *P5 L17-19*

   *I would remove these 2 sentences. They confuse more than explain*

   We decided to maintain this paragraph because we think that it is important to point out that FRF must be interpreted as an average value.

6. *P6 L25*

   *Maybe more simple: "This tracer is linearly increasing in time in the lowest model layer"*

   We changed the sentence according to the referee.

7. *P7 L25 You should discuss different lifetimes of your idealized tracers earlier in the text, e.g. at the end of section 3.1 (after L21). It would make the following text easier to understand. Maybe you simply shift the lifetime discussion to this place.*

   As suggested by the referee, we added a paragraph on the stratospheric lifetimes of the tracers in Sect. 3.1. The lifetimes are taken from the SPARC Lifetimes Report (2013), Chapter 5. In Sect. 4 we still discuss the dependence of FRF on the lifetime of the species and therein we now refer to Sect. 3.1. *"
[revised manuscript text omitted]